

# Ocean acidification decreases plankton respiration: evidence from a mesocosm experiment

K. Spilling[1,2], A. J. Paul[3], N. Virkkala[2], T. Hastings[4], S. Lischka[3], A. Stuhr[3], R. Bermudez[3,5], J. Czerny[3], T. Boxhammer[3], K. G. Schulz[6], A. Ludwig[3], and U. Riebesell[3]

[1]Marine Research Centre, Finnish Environment Institute, P.O. Box 140, 00251 Helsinki, Finland
[2]Tvärminne Zoological Station, University of Helsinki, J. A. Palménin tie 260, 10900 Hanko, Finland
[3]GEOMAR Helmholtz Centre for Ocean Research Kiel, Düsternbrooker Weg 20, 24105 Kiel, Germany
[4]Department of Biology, University of Portsmouth, University House, Winston Churchill Avenue, Portsmouth PO1 2UP, UK
[5]Facultad de Ingenierí a Marítima, Ciencias Biológicas, Oceánicas y Recursos Naturales. ESPOL, Escuela Superior Politécnica del Litoral, Guayaquil, Ecuador
[6]Centre for Coastal Biogeochemistry, Southern Cross University, Military Road, East Lismore, NSW 2480, Australia



Received: 27 November 2015 – Accepted: 3 December 2015 – Published: 15 January 2016

Correspondence to: K. Spilling (kristian.spilling@environment.fi)

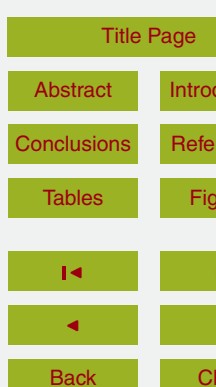

Discussion Paper | Discussion Paper | Discussion Paper | Discussion Paper | Discussion Paper |

**BGD**

doi:10.5194/bg-2015-608

**Ocean acidification decreases respiration**

K. Spilling et al.

## Abstract

Anthropogenic carbon dioxide ($CO_2$) emissions are reducing the pH in the world's oceans. The plankton community is a key component driving biogeochemical fluxes, and the effect of increased $CO_2$ on plankton is critical for understanding the ramifica-

5 tions of ocean acidification on global carbon fluxes. We determined the plankton community composition and measured primary production, respiration rates and carbon export (defined here as carbon sinking out of a shallow, coastal area) during an ocean acidification experiment. Mesocosms ($\sim 55\,m^3$) were set up in the Baltic Sea with a gradient of $CO_2$ levels initially ranging from ambient ($\sim 240\,\mu atm$), used as control, to high

$CO_2$ (up to $\sim 1330\,\mu atm$). The phytoplankton community was dominated by dinoflagellates, diatoms, cyanobacteria and chlorophytes, and the zooplankton community by protozoans, heterotrophic dinoflagellates and cladocerans. The plankton community composition was relatively homogenous between treatments. Community respiration rates were lower at high $CO_2$ levels. The carbon-normalized respiration was approx-

imately 40 % lower in the high $CO_2$ environment compared with the controls during the latter phase of the experiment. We did not, however, detect any effect of increased $CO_2$ on primary production. This could be due to measurement uncertainty, as the measured total particular carbon (TPC) and combined results presented in this special issue suggest that the reduced respiration rate translated into higher net carbon

fixation. The percent carbon derived from microscopy counts (both phyto- and zooplankton), of the measured total particular carbon (TPC) decreased from $\sim 26$ % at t0 to $\sim 8$ % at t31, probably driven by a shift towards smaller plankton ($< 4\,\mu m$) not enumerated by microscopy. Our results suggest that reduced respiration lead to increased net carbon fixation at high $CO_2$. However, the increased primary production did not

translate into increased carbon export, and did consequently not work as a negative feedback mechanism for increasing atmospheric $CO_2$ concentration.

**BGD**

doi:10.5194/bg-2015-608

**Ocean acidification decreases respiration**

K. Spilling et al.

Discussion Paper | Discussion Paper | Discussion Paper | Discussion Paper

Discussion Paper | Discussion Paper | Discussion Paper | Discussion Paper |

**BGD**

doi:10.5194/bg-2015-608

**Ocean acidification decreases respiration**

K. Spilling et al.

# 1 Introduction

The ocean is a large sink of carbon dioxide ($CO_2$) and absorbs around 25 % of annual anthropogenic $CO_2$ emissions (Le Quéré et al., 2009). $CO_2$ is a weak acid when dissolved in water, and the increasing global atmospheric $CO_2$ concentration has re-
5 duced the average pH in the ocean by approximately 0.1 since the start of the Industrial Revolution (Orr, 2011). This pH reduction, with a concurrent increase in dissolved inorganic carbon, is called ocean acidification. Following the same trajectory, the pH could decline further by as much as 0.7 by 2300 (Zeebe et al., 2008).

The topic of ocean acidification has received a lot of attention over the past decade.
There is a relatively good understanding of the rate of change and the effects on the ocean's carbon chemistry (Zeebe and Ridgwell, 2011). There are also a range of studies documenting the effects of decreasing pH on marine life, but the effect studied is often species or ecosystem specific and based on short term perturbation experiments (Riebesell and Tortell, 2011). There are still a lot of uncertainties as to what effect
ocean acidification has on biological processes.

The key driving force in marine biogeochemical element cycling is the planktonic community that occupies the sunlit surface of the ocean. Primary producers use the energy from sunlight to take up $CO_2$ and fix carbon into organic compounds. Respiration is the opposite process where organic carbon is oxidized providing energy and releas-
20 ing $CO_2$. This takes place at all trophic levels, from bacteria through to zooplankton, fish and marine mammals. At steady state, production and respiration are balanced. On a global scale, there is presently a surplus of organic matter being produced in the upper ocean through photosynthesis. The extra organic carbon is exported out of the surface layers to the deep ocean where it is sequestered for the foreseeable future,
a process referred to as the biological carbon pump. (Volk and Hoffert, 1985; Siegenthaler and Sarmiento, 1993; Ducklow et al., 2001). In the case of coastal seas, part of the carbon is buried at the sea floor (Dunne et al., 2007).

The greater the difference between primary production and respiration, the more carbon can potentially be exported, and ocean acidification has the potential to affect this balance. Generally, more $CO_2$ stimulates photosynthetic carbon fixation, as $CO_2$ becomes more readily available for the key photosynthetic enzyme RubisCO (Falkowski and Raven, 2013), however, increased primary production at high $CO_2$ concentration is not always recorded (Sobrino et al., 2014) and the response is variable between different taxa (Mackey et al., 2015). In cases where additional carbon is fixed, it may be excreted as dissolved organic carbon (DOC), providing carbon for bacterial growth, and also increasing bacterial respiration (Grossart et al., 2006; Piontek et al., 2010). Changes in pH might also directly affect both primary production (Spilling, 2007) and respiration (Smith and Raven, 1979).

The Baltic Sea is an almost landlocked sea with low alkalinity (Beldowski et al., 2010), and is thus particularly susceptible to variation in seawater pH. Because of the reduced water exchange with the North Atlantic and the large drainage area (population $\sim$ 80 million), it is also subjected to a range of other environmental pressures, in particular eutrophication i.e. increased nutrient inputs from human activities. Eutrophication has led to increased primary production and chlorophyll *a* (Chl *a*) biomass over the past decades in the Gulf of Finland (Raateoja et al., 2005), benefitting chrysophytes, chlorophytes and cyanobacteria, (Suikkanen et al., 2007). Dense blooms of diazotroph cyanobacteria are common in the summer, which further aggravates the eutrophication problem as nitrogen fixation introduces substantial amounts of new nitrogen into the system (Savchuk, 2005). The effect of ocean acidification on this type of system is largely unexplored. In order to investigate the effect of increased $CO_2$ (and lower pH) on primary production and total plankton respiration in the pelagic zone, we measured carbon fixation, oxygen consumption and export/sedimentation rates during a $CO_2$-manipulation study set up in the Gulf of Finland, Baltic Sea (further references within this special issue).

Discussion Paper | Discussion Paper | Discussion Paper | Discussion Paper | Discussion Paper |

**[BGD]**

doi:10.5194/bg-2015-608

**Ocean acidification decreases respiration**

K. Spilling et al.

Discussion Paper | Discussion Paper | Discussion Paper | Discussion Paper |

**BGD**

doi:10.5194/bg-2015-608

**Ocean acidification decreases respiration**

K. Spilling et al.

## 2   Materials and methods

### 2.1   Experimental set-up

Six pelagic mesocosms (approximately 55 m$^3$) were moored at Storfjärden, SW coast of Finland (59°51.5′ N; 23°15.5′ E) on 12 June 2012. The water depth at the mooring site is approximately 30 m and the mesocosms extended from the surface down to 19 m depth. A more detailed description of the mesocosm bags and the experimental area can be found in Paul et al. (2015).

On t5 (17 June 2012, 5 days before the first $CO_2$ enrichment), the mesocosms were bubbled with compressed air to break down any existing pycnocline and ensure homogeneous water mass distribution. Different $CO_2$ concentrations in the bags were achieved by adding filtered (50 μm), $CO_2$-saturated seawater. This was done stepwise in four separate additions to reduce the shock of rapid change in pH on the plankton community. The first addition took place after sampling on t0, thus t1 was the first day with a $CO_2$ treatment. The $CO_2$ enriched water was evenly distributed over the upper 17 m using a specially designed distribution device i.e. "spider" (Riebesell et al., 2013). Two controls and four treatment mesocosms were used. Filtered water (with ambient $CO_2$ concentration) was added to the control mesocosms at the time when $CO_2$ was manipulated in the treatment mesocosms. The $CO_2$ fugacity gradient on t4, after the four additions, ranged from ambient ($\sim$ 240 μatm $f\mathrm{CO_2}$) in the two control mesocosms (M1 and M5), up to $\sim$ 1650 μatm $f\mathrm{CO_2}$, but we used the average $f\mathrm{CO_2}$ throughout the relevant part of this experiment (from t1–t31) to denote the different treatments: 346 (M1), 348 (M5), 494 (M7), 868 (M6), 1075 (M3) and 1333 (M8) μatm $f\mathrm{CO_2}$. On t15, additional $CO_2$ enriched seawater was added to the upper 7 m in the same manner as the initial enrichment to counteract outgassing of $CO_2$. The mesocosm bags were regularly cleaned to prevent wall growth. A more detailed description of the treatment and cleaning can be found in Paul et al. (2015).

Mesocosm sampling was carried out every day (or every second day for some parameters) in the morning. Two different depth-integrated water samples (0–10 and 0–

Title Page

Abstract | Introduction

Conclusions | References

Tables | Figures

◄◄ | ►►

◄ | ►

Back | Close

Full Screen / Esc

Interactive Discussion

**[BGD]**

doi:10.5194/bg-2015-608

**Ocean acidification decreases respiration**

K. Spilling et al.

17 m) were taken using integrating water samplers (IWS, HYDRO-BIOS, Kiel). The water was collected into plastic carboys (10 L) and brought to the laboratory for sub-sampling and subsequent analysis of plankton community composition, carbon concentration and for respiration and primary production incubations. Sub-samples for primary production and respiration measurements were treated and stored minimizing the contact with the air in order to prevent any gas exchange.

Settling particles were quantitatively collected in the sediment traps at the bottom end of the mesocosm units at 19 m water depth. Every 48 h the accumulated material was vacuum pumped through a silicon tube to the sea surface and transferred into 5 L glass bottles for transportation to the laboratory. For a more detailed description of the sampling procedure and sample processing of the sediment see Boxhammer et al. (2015).

## 2.2 Phytoplankton community

Phytoplankton cells were counted in 50 mL sub-samples, which were fixed with acidic Lugol's iodine solution (1 % final concentration) with an inverted microscope (ZEISS Axiovert 100) after Utermöhl (1958). The cells > 20 μm were counted either from half of the chamber at 100-fold or on 3 to 4 stripes at 200-fold magnification. Filamentous cyanobacteria were counted in 50 μm length units. Cells 12–20 μm were counted at half of the chamber at 200-fold magnification, and cells 4–12 μm were counted at 400-fold magnification on two radial strips. The phytoplankton counts of the smaller size classes (< 20 μm) stopped on t29, and these results have been used together with the t31 results for larger (> 20 μm) phytoplankton as the end date of the experiment. Further details can be found in Bermúdez et al. (2015)

Phytoplankton, heterotrophic dinoflagellates and protozoa were identified with the help of Tomas (1997); Hoppenrath et al. (2009) and Kraberg et al. (2010). Biovolumes of counted plankton cells were calculated according to Olenina (2006) and converted to cellular organic carbon quotas by the equations of Menden-Deuer and Lessard (2000).

## 2.3 Microzooplankton community

Ciliates were enumerated from 50 mL sub-samples every second day with a Zeiss Axiovert 100 inverted microscope (Utermöhl 1958) at 200× magnification. At high cell numbers (> 400 cells), half the bottom plate area was counted. If less than 400 cells were found in the first half of the bottom plate area, the entire chamber was counted. Rare species were counted on the whole bottom plate. Ciliates were identified to the lowest possible taxonomic level (genus/species) according to Setälä et al. (1992); Telesh et al. (2009). For more details see Lischka et al. (2015) in this issue.

## 2.4 Mesozooplankton community

Mesozooplankton samples were collected by net hauls from 17 m depth with an Apstein net of 17 cm diameter and 100 µm mesh size. After closing of the mesocosm bags, mesozooplankton samples were taken prior to the $CO_2$ addition on t0 and at t17 and t31 (there were also other sampling days for zooplankton but these are not included here). Samples were preserved in 70 % ethanol. Zooplankton was counted assuming 100 % filtering efficiency of the net. The samples were divided with a Folsom plankton splitter (1 : 2, 1 : 4, 1 : 8, 1 : 16, and 1 : 32) and aliquots were counted using a WILD M3B stereomicroscope. Abundant species/taxa were enumerated from sub-samples (> 30 individuals in an aliquot) while less abundant and rare species/taxa were counted from the whole sample. For more details on mesozooplankton collection, processing and species determination, see Lischka et al. (2015). Carbon biomass (CB) in µmol C L$^{-1}$ was calculated using the displacement volume (DV) and the equation of Wiebe (1988):

$$(\log DV + 1.429)/0.82 = \log CB \tag{1}$$

Discussion Paper | Discussion Paper | Discussion Paper | Discussion Paper |

**BGD**

doi:10.5194/bg-2015-608

**Ocean acidification decreases respiration**

K. Spilling et al.

## 2.5 Total particulate carbon

Samples for total particulate carbon (TPC) measurements were sub-sampled from 10 L carboys and filtered onto GF/F filters (Whatman, nominal pore size of 0.7 µm, diameter = 25 mm) under reduced vacuum (< 200 mbar). Sampling for TPC occurred every
2nd day from t3 until the end of the experiment. Filters were stored in glass petri dishes at −20 °C directly after filtration until preparation of samples for analyses. Petri dishes and filters were combusted at 450 °C for 6 h before use.

Samples were analyzed for total particulate carbon (organic + inorganic) as no acidifying step was made to remove particulate inorganic carbon. Filters were dried at 60 °C
and packed into tin capsules and stored in a desiccator until analysis on an elemental analyzer (EuroEA) as described by (Sharp, 1974).

The particles collected from the sediment traps were allowed to settle down in the sampling flasks at in-situ temperature before separation of supernatant and the dense particle suspension at the bottom. TPC content of the supernatant was analysed from
10–50 mL sub-samples as described above for water column measurements. The dense particle suspension was concentrated by centrifugation, then freeze-dried and ground to a very fine powder of homogeneous composition. From this material, small sub-samples of 1–2 mg were transferred into tin capsules and TPC content was analysed analogue to the supernatant and water column samples. Vertical carbon flux was
calculated from the two measurements and is given as the daily amount of TPC (mmol) collected in the sediment traps per square meter of mesocosm surface area (3.142 m$^2$).

## 2.6 Dissolved inorganic carbon

Samples for dissolved inorganic carbon (DIC) were gently pressure-filtered (Saarstedt Filtropur 0.2 µm) before measurements to remove all particulates. DIC concentrations
were determined by infrared absorption (LICOR LI-7000 on an AIRICA system, Marianda). Four (2 mL) replicates were measured, and the final DIC concentration was calculated from the mean of the three most consistent samples.

Discussion Paper | Discussion Paper | Discussion Paper | Discussion Paper | Discussion Paper |

**BGD**

doi:10.5194/bg-2015-608

**Ocean acidification decreases respiration**

K. Spilling et al.

## 2.7 Plankton community respiration

Samples for respiration rate measurements were subsampled from the depth integrated sample from the entire water column (0–17 m). Oxygen was measured using a fiber optic dipping probe (PreSens, Fibox 3), which was calibrated against anoxic (0 % $O_2$, obtained by adding sodium dithionite) and air saturated water (obtained by bubbling sampled water with air for 5 min followed by 15 min of stirring with a magnetic stirrer). The final $O_2$ concentration was calculated using the Fibox 3 software including temperature compensation.

We filled three replicate 120 mL $O_2$ bottles (without headspace) for each mesocosm. After the initial $O_2$ determination, the bottles were put in a dark, temperature controlled room, set to the ambient water temperature as determined by CTD profiles (0–18 m). The $O_2$ concentration was determined again after an incubation period of 48 h, and the oxygen consumption (i.e. respiration rate) was calculated from the difference between the $O_2$ concentration before and after the incubation period. Respiration rates were measured every day t-3 to t31, with the exception of days: t2 and t14 because of technical problems.

## 2.8 Primary production

Primary production was measured using radio labeled $NaH^{14}CO_3$ (Steeman-Nielsen, 1952) from the 0–10 m depth integrated sample. The rational for using the upper (0–10 m) part of the mesocosm was the low light penetration depth, and 0–10 m was representative of the euphotic zone. The water was gently filled into 12 small (8 mL) scintillation vials per mesocosm and 10 µL of $^{14}C$ bicarbonate solution (DHI Lab; 20 µCi mL$^{-1}$), was added. The vials were filled completely and after adding the cap there was only a very small (2–3 mm) air bubble remaining corresponding to $\sim 0.1$ % of total volume.

Duplicate samples for each mesocosm were incubated just below the surface and at 2, 4, 6, 8 and 10 m depths for 24 h on small incubation platforms moored next to the

**BGD**

doi:10.5194/bg-2015-608

**Ocean acidification decreases respiration**

K. Spilling et al.

mesocosms (Fig. S1). In addition, a dark incubation (vials covered with aluminium foil) was incubated at the same location at 11 m depth.

After incubation, 3 mL of the sample was removed from each vial and acidified with 100 μL 1 mol L$^{-1}$ HCl, and left without a lid for 24 h to ensure removal of remaining inorganic $^{14}$C. Four mL of scintillation cocktail (Instagel Plus, Perkin Elmer) was added, and the radioactivity was determined using a scintillation counter (Wallac 1414, Perkin Elmer). Primary production was calculated knowing the $^{14}$C incorporation (with dark values subtracted) and the fraction of the $^{14}$C addition to the total inorganic carbon pool according to Gargas (1975). The primary production incubations were set up at the same time as the respiration incubations, but here we missed measurements for two periods: t1–t3 and t6–t8, due to loss of the incubation platform.

## 2.9 Data treatment

The average of the three respiration bottles was used to calculate the respiration rate. There were two days without measurements: t2 and t14 and for these days we estimated the respiration rate by using the average of the day before and after this day. TPC was measured only every second day, therefore for the days without TPC measurements we normalized respiration to average TPC from the day before and the day after the respiration measurement.

The cumulative respiration was calculated by adding the total oxygen consumption for each day. When evaluating the data, there were two clear periods emerging from the experiment: the initial period t0–t16 (Phase I) and period from t17–t31 (Phase II) when the effect of the $CO_2$ addition was more evident. This division was also seen in e.g. Chl $a$ and temperature (Paul et al., 2015). Using the respiration data from Phase II we calculated the average respiration for each treatment by linear regression. From the linear regression, the standard error (SE) from the residuals and the coefficient of determination ($R^2$) were calculated, in addition to a statistical test comparing the linear regression with a flat line, using Sigma Plot software.

Discussion Paper | Discussion Paper | Discussion Paper | Discussion Paper |

**BGD**

doi:10.5194/bg-2015-608

**Ocean acidification decreases respiration**

K. Spilling et al.

The areal primary production was calculated based on a simple linear model of the production measurements from the different depths (Fig. S2 in the Supplement). The cumulative primary production was carried out similar to respiration, but as the two missing periods were > 1 day, we did not estimate missing values, and the final cumu-
lative production is therefore a slight underestimate (missing 6 days of production). We normalized the production data to the TPC in the euphotic zone, defined by the areal production model (Fig. S2).

## 3   Results

### 3.1   Phytoplankton community composition

The phytoplankton community in the mesocosms was dominated by dinoflagellates, cyanobacteria, diatoms, chrysophytes and chlorophytes at the start of the experiment (Fig. 1). The two latter groups consisted almost exclusively of small cells (< 20 μm). There was an initial increase in phytoplankton biomass from an average of 3 μmol C L$^{-1}$ to a maximum of ∼ 4.1 μmol C L$^{-1}$ in the two controls (M1 and M5), but at the end of
Phase I (t0–t16) the biomass had declined and at t17 it ranged between 3.2 to 3.5 μmol C L$^{-1}$. During Phase I, large (> 20 μm) diatoms decreased in abundance and eugleno-phytes increased from a negligible group initially (0.5 % of the biomass) to constitut-ing 15–25 % of the autotrophic biomass at t17. It was, however, the small (< 20 μm) phytoplankton cells (small diatoms, chrysophytes and chlorophytes) that made up the
majority (70–80 %) of the counted autotroph biomass during Phase I.

During Phase II (t17–t31), there was a decline in phytoplankton biomass to 0.5–1 μmol C L$^{-1}$ and at t31 dinoflagellates had become the dominating group in all treat-ments except at the highest $CO_2$ level. Cyanobacteria and chlorophytes were also abundant and the dominating groups in the highest $CO_2$. There was no consistent
difference between phytoplankton communities in the different $CO_2$ treatments, but dinoflagellate abundance was lower in the highest $CO_2$ treatment (M8), and conse-

**BGD**

doi:10.5194/bg-2015-608

**Ocean acidification decreases respiration**

K. Spilling et al.

Discussion Paper | Discussion Paper | Discussion Paper | Discussion Paper |

quently the total phytoplankton biomass was lower in this treatment at t31. The relative increase of large dinoflagellates decreased the contribution of the smaller autotroph size class (4–20 μm) to 40–60 % of the counted phytoplankton biomass at t31.

## 3.2 Zooplankton community composition

Protozoans, ciliates and heterotrophic dinoflagellates dominated the microzooplankton and constituted a major part (2.8 μmol C L$^{-1}$) of the whole zooplankton community at the start of the experiment (Fig. 2). Protozoans, dominated by the choanoflagellate *Calliacantha natans*, decreased from the initial high concentrations during Phase I, in particular in the M1 control bag. The photosynthesizing, *Myrionecta rubra* made up approximately half of the ciliate biomass at t0, but both this species and the total biomass of ciliates decreased during Phase I. The biomass of heterotrophic dinoflagellates was relatively stable throughout Phase I, but started to decrease during Phase II.

The mesozooplankton community was initially dominated by copepods, cladocerans and rotifers (Fig. 2). The average initial biomass was 0.05 μmol C L$^{-1}$ and increased to 0.13 μmol C L$^{-1}$ at t17. During Phase I, copepods became the dominating group with > 50 % of the mesozooplankton biomass. In Phase II of the experiment, mesozooplankton biomass increased and was on average 0.27 μmol C L$^{-1}$ at t31. This was caused by an increase in cladocerans, mainly *Bosmina* sp., whereas copepod biomass was more constant over the course of the experiment. The population peak of *Bosmina* sp. had slightly different timing in the different mesocosms but was higher in the mesocosms with added $CO_2$, except for the highest $CO_2$ addition (M8).

## 3.3 Total particulate carbon and export of carbon

Average TPC was 22.5 μmol C L$^{-1}$ at the beginning of the experiment and after an initial increase to 32 μmol C L$^{-1}$ it decreased to 19.2 μmol C L$^{-1}$ at t17 (Fig. 3). In the beginning of Phase II it was relatively stable and with no clear effect of $CO_2$ treatment, but at the end of the study period (t31) there was more TPC in the higher $CO_2$

Discussion Paper | Discussion Paper | Discussion Paper | Discussion Paper |

**BGD**

doi:10.5194/bg-2015-608

**Ocean acidification decreases respiration**

K. Spilling et al.

treatments, and the increase in TPC during Phase II was highest in the $CO_2$ additions (Table 1). At t31 the average TPC was 19.9 µmol C L$^{-1}$, ranging from 18.9 ± 0.6 (SE) µmol C L$^{-1}$ in the controls to 22.1 µmol C L$^{-1}$ in the highest $CO_2$ treatment.

The carbon accounted for by biologically active organisms counted in the microscope (phytoplankton and zooplankton) was initially 26 % of the TPC. At t17 and t31 this percentage decreased to $\sim$ 20 and $\sim$ 8 % respectively.

The export of carbon, defined here as carbon settling out of the mesocosms, decreased during the experiment and there was no effect of $CO_2$ concentration. The average export of TPC was in the range of 6.1–7.4 mmol C m$^{-2}$ d$^{-1}$ during Phase I (Table 1). This decreased to 2.5–3.3 mmol C m$^{-2}$ d$^{-1}$ during Phase II.

## 3.4 Primary production and respiration

There was no clear effect of $CO_2$ addition on primary production (Fig. 4). There were relatively large daily variations in depth-integrated primary production depending on the light environment, and days with clear skies and more light increased carbon fixation. One of the control bags (M1) had clearly lower primary production from the very start of the experiment, and this was evident even before the initiation of the $CO_2$ addition (Fig. 4). The average production during the whole experiment was 3.67 ± 0.42 (SE) mmol C m$^{-2}$ d$^{-1}$ in M1, and for all other bags 10.5 ± 0.67 (SE) mmol C m$^{-2}$ d$^{-1}$. Production on clear, sunny days was (except for M1) approximately 25 mmol C m$^{-2}$ d$^{-1}$. The general pattern in areal primary production was similar to TPC-normalized production (Table 1). Cumulative production values in mol C m$^{-2}$ are presented in the Supplement (Fig. S3).

The respiration rate was higher in the ambient than the high $CO_2$ treatments (Fig. 5). In one of the two controls (M1), the respiration rate was clearly higher compared to all other treatments from the beginning of the experiment. The respiration rate in the other control (M5) increased approximately two weeks later than the $CO_2$ treatments. After t17, the mesocosm with highest $CO_2$ concentration (average of 1333 µatm

Discussion Paper | Discussion Paper | Discussion Paper | Discussion Paper |

**BGD**

doi:10.5194/bg-2015-608

**Ocean acidification decreases respiration**

K. Spilling et al.

Discussion Paper | Discussion Paper | Discussion Paper | Discussion Paper |

**BGD**

doi:10.5194/bg-2015-608

**Ocean acidification decreases respiration**

K. Spilling et al.



$f\mathrm{CO_2}$) started to have lower cumulative respiration compared to those with intermediate $\mathrm{CO_2}$ levels (494–1075 µatm $f\mathrm{CO_2}$). After another week ($\sim$ t27), differences between the intermediate $\mathrm{CO_2}$ treatments became apparent. At the end of Phase II (t20–t31), there was a 40 % difference in respiration rate between the lowest and highest $f\mathrm{CO_2}$ treatments ($p = 0.02$; Fig. 6). The volumetric respiration during Phase II was 7.6 and 7.1 µmol $\mathrm{O_2}$ $\mathrm{L^{-1}}$ $\mathrm{d^{-1}}$ for the two controls, and 4.7–5.7 µmol $\mathrm{O_2}$ $\mathrm{L^{-1}}$ $\mathrm{d^{-1}}$ for the $\mathrm{CO_2}$ treatment mesocosms. Outside the mesocosms, at ambient $\mathrm{CO_2}$ concentration (average of 343 µatm $f\mathrm{CO_2}$ but with larger variability than inside the mesocosms), the carbon normalized respiration rate was lower than inside the mesocosms and the cumulative, carbon-normalized respiration was approximately half of that measured in the control bags at the end of the experiment (Fig. 5). The general pattern of lower respiration rates at high $\mathrm{CO_2}$ concentration was the same without normalization to TPC (Table 1, Fig. S4).

## 4 Discussion

### 4.1 Plankton community

The particulate and dissolved standing stocks during this experiment are presented in Paul et al. (2015). In the initial Phase I of the experiment the Chl $a$ concentration was relatively high ($\sim$ 2 µg Chl $a$ $\mathrm{L^{-1}}$), but started to decrease during Phase II, and reached $\sim$ 1 µg Chl $a$ $\mathrm{L^{-1}}$ at t31 in all of the treatments. During this transition there was a shift in the plankton community with decreasing phytoplankton and microzooplankton, and increasing abundance of mesozooplankton, primarily cladocerans (Figs. 1 and 2).

The phytoplankton community composition was dominated by common species in the area (Hällfors, 2004). In the latter part (Phase II), the relative dominance by dinoflagellates was mainly due to reduction in biomass of the other groups, with the exception of the highest $\mathrm{CO_2}$ concentration where also the dinoflagellates decreased in abundance. Dinoflagellates are generally favored in low turbulence (Margalef, 1978;

Smayda and Reynolds, 2001), and were probably benefitting from the relative stable conditions within the mesocosms. Blooms of filamentous cyanobacteria do occur in the area, but did not develop within the mesocosms. The relatively low temperature (mostly < 15 °C; Paul et al., 2015) could be a reason for that (Kanoshina et al., 2003).

Protozoans, ciliates and heterotrophic dinoflagellates dominated the microzooplankton, and *Myrionecta rubra* initially made up a large proportion of the ciliates. *M. rubra* can be regarded as mixotropic and would also have contributed to the carbon fixation (Johnson et al., 2006). Copepods and cladocerans initially dominated the mesozooplankton and during Phase II cladocerans became the dominant mesozooplankton group. Cladocerans are typically predominant in freshwater but in the brackish Baltic Sea they can be common, in particular when stability in the water column is high (Viitasalo et al., 1995).

The combined phyto- and zooplankton carbon derived from microscope counts decreased during the experiment. TPC did not decrease to the same extent, and the percentage microscope-derived carbon of TPC decreased from 26 % at t0 to only ~ 8 % of the measured TPC at t31. These numbers are not directly comparable, as detritus, i.e. non-living carbon particles, are included in TPC. However, any large aggregates sink rapidly and are not expected to have contributed much to the TPC. The reduction of microscopy-derived carbon to TPC indicate rather increasing importance of smaller size classes ($< 4\,\mu m$), not enumerated by the microscope counts. This conclusion is also supported by flow cytometer data from this experiment (Crawfurd et al., 2015), increasing uptake of $PO_4$ by the $< 3\,\mu m$ fraction (Nausch et al., 2015) and the increasing proportion of the smallest ($< 2\,\mu m$) size class of Chl *a* (Paul et al., 2015).

## 4.2  Primary production and respiration

Primary production and respiration rates were comparable to values obtained under similar conditions in the area (Kivi et al., 1993). There are relatively few records of respiration, but the measured respiration rates in the control bags were similar to the average respiration rate obtained for a range of coastal waters of $7.4\pm0.54\,\mathrm{mmol\,O_2\,m^{-3}\,d^{-1}}$

Discussion Paper | Discussion Paper | Discussion Paper | Discussion Paper |

**[BGD]**

doi:10.5194/bg-2015-608

**Ocean acidification decreases respiration**

K. Spilling et al.

**[BGD]**

doi:10.5194/bg-2015-608

**Ocean acidification decreases respiration**

K. Spilling et al.

($n$ = 323) (Robinson and Williams, 2005). The incubation period we used for primary production measurements (24 h) provides production rates close to net production (Marra, 2009). The oxidation of organic carbon (C respiration) can be estimated from $O_2$ consumption using the respiratory quotient (RQ), which is defined as mole $CO_2$
produced per mole $O_2$ consumed. The RQ depends on what form the oxidized carbon has (e.g. carbohydrates, lipids or proteins), but for carbohydrate oxidation, the RQ is $\sim$ 1 (Buchanan et al., 2000). Using an RQ of 1, the respired carbon was approximately an order of magnitude higher than the net production (Table 1).

  The higher respiration and lower production in the M1 control bag was probably con-
nected, i.e. higher respiration lead to lower net carbon fixation, however, the reason for the M1 bag being very different from the very start is not clear. Most of the other parameters were similar in the M1 bag compared to the rest (Paul et al., 2015), but there was some indication of difference in community. In particular, protozoans were lower in the M1 bag compared with the rest of the mesocosms throughout the experiment.
However, judging from the development in carbon pools (Paul et al., 2015) and fluxes in the system (Spilling et al., 2015), the NPP measurements for the M1 bag is likely an underestimate. Bacterial production during Phase II was highest in the ambient $CO_2$, in particular in M1 (Hornick, 2015), and could partly be the reason for the elevated respiration rate in this mesocosm bag.

## 4.3  Effect of $CO_2$ on the balance between respiration and carbon fixation

Increased $CO_2$ concentration has increased carbon fixation in some studies (Egge et al., 2009; Engel et al., 2013). This was not observed in this study, but the higher Chl $a$, TPC and DOC in the high $CO_2$ treatments at the end of the experiment (Paul et al., 2015) could have been caused by the lower respiration rate in the highest $CO_2$
enriched mesocosms, rather than increased primary production. Bacterial production was higher in the low $CO_2$ after t20 during this experiment (Hornick, 2015), which fits with the higher respiration rate at ambient $CO_2$ concentration. The biomass of the smallest plankton size fraction ($<$ 4 μm, not counted by microscope) increased in rel-

Discussion Paper | Discussion Paper | Discussion Paper | Discussion Paper |

ative importance with $CO_2$ addition in the latter part of the experiment, in particular pico-eukaryotes (Crawfurd et al., 2015), and seems to have benefitted most by elevated $CO_2$ concentration, similar to findings in the Arctic (Brussaard et al., 2013).

This study is, to our knowledge, the first one describing reduced respiration rates with ocean acidification on a plankton community scale. There are relatively few measurements of community respiration in ocean acidification experiments, and existing studies have revealed no specific responses in respiration (Egge et al., 2009; Tanaka et al., 2013; Mercado et al., 2014). Some of these studies have been relatively short (< 2 weeks) compared to the current study. Our results revealed a $CO_2$ effect only two weeks into the experiment, suggesting that potential effects may have been present but remained below the detection limits.

The effect of increasing $CO_2$ concentration on respiration has mostly been documented for single species. For example, the copepod *Centropages tenuiremis* (Li and Gao, 2012) and the diatom *Phaeodactylum tricornutum* (Wu et al., 2010) exhibited increased respiration rates in a high $CO_2$ environment ($\geq 1000\,\mu$atm $f\mathrm{CO}_2$), contrary to our findings. However, these types of studies have revealed different responses even when comparing different populations of the same species (Thor and Oliva, 2015), and any interpolation from single-species, laboratory-studies should be carried out with great caution. The larger-scale mesocosm approach taken here has the advantage that the whole plankton community and interacting effects between different components of the food web are included.

In higher plants, it is known that elevated $CO_2$ decreases mitochondrial respiration in the foliage (Puhe and Ulrich, 2001). In their review, Drake et al. (1999) outlined two $CO_2$ effects on respiration: an immediate, reversible effect and a longer term, irreversible effect, both decreasing respiration in a high $CO_2$ environment. In our study it was only a longer term effect that was observed. It is not known what cause this reduced respiration in plant foliage, but Amthor (1991) pointed out seven potential mechanisms for how changes in the $CO_2$ concentration could reduce plant respiration, for example by affecting respiratory enzymes. A doubling of present day $CO_2$ concentration could decrease

Discussion Paper | Discussion Paper | Discussion Paper | Discussion Paper |

**BGD**

doi:10.5194/bg-2015-608

**Ocean acidification decreases respiration**

K. Spilling et al.

foliage respiration rate by 15 to 30 % (Drake et al., 1999; Puhe and Ulrich, 2001), but other parts e.g. root system are projected to increase respiration so the net effect of elevated $CO_2$ on plant respiration is uncertain (Puhe and Ulrich, 2001). Phytoplankton lack any specialized structures like root system and may consequently function more like plant foliage, but this is an underexplored research avenue that deserves further study.

The intracellular pH can be highly variable between different cellular compartments and organelles, but in the cytosol the pH is normally close to neutral (pH $\sim$ 7.0), and is to a large extent independent of the external pH (Roos and Boron, 1981). In plants, animals and also bacteria, there is a complex set of pH regulatory mechanisms that is fundamentally controlled by physiological processes such as membrane transport of $H^+$ or $OH^-$ and intracellular metabolism (Smith and Raven, 1979; Kurkdjian and Guern, 1989). Internal pH regulation can be a considerable part of baseline respiration (Pörtner et al., 2000). With ocean acidification, the external pH becomes closer to the intracellular pH, and might reduce the metabolic cost (respiration) related to internal pH regulation. The intracellular pH regulation works similar in single-cell or multi-cell organisms, but judging from the importance of the smallest size class in this study, bacterial and picophytoplankton community (Crawfurd et al., 2015) and bacterial production (Hornick, 2015), the decreased respiration at higher $CO_2$ concentration was probably mostly due to picoplankton.

## 4.4 Interacting effects and community composition

It is evident from our measurements outside the mesocosm bags that plankton physiology and community composition can have a big impact on both primary production and respiration. The plankton community was relatively uniform across all mesocosm bags. Unfortunately we do not have any community data from outside the mesocosm bags, but the amplitude of Chl $a$ dynamics was different, with an upwelling event leading to a doubling of the Chl $a$ concentration ($\sim$ 5 µg Chl $a$ L$^{-1}$) around t17 (Paul et al., 2015). This suggests a different availability of inorganic nutrients and different plank-

## BGD

doi:10.5194/bg-2015-608

**Ocean acidification decreases respiration**

K. Spilling et al.

ton community as other environmental parameters such as light and temperature were similar both inside and outside the mesocosm bags. The carbon-normalized respiration rate outside the mesocosm bags (with ambient $f\mathrm{CO_2}$) was approximately half of the respiration rates in the controls with the same average $f\mathrm{CO_2}$, and also absolute
respiration was clearly lower during Phase II, when nitrate was depleted inside the bags and plankton biomass was decreasing. However, the $f\mathrm{CO_2}$ was more variable outside the mesocosm bags compared with the control bags (although their averages were similar), and the $f\mathrm{CO_2}$ increased throughout Phase II outside the bags to approximately 700 µatm by t31 (Paul et al., 2015). This could have influenced the carbon
normalized respiration, which started to deviate outside the bags during Phase II, but it could also have been interacting effects of different environmental changes (different nutrient dynamics) leading to this lower respiration rate. An often overlooked aspect is the importance of plankton community composition, which can be more important than changes in external factors (Verity and Smetacek, 1996; Eggers et al., 2014).
Bacterial production (Grossart et al., 2006) and bacterial degradation of polysaccharides (Piontek et al., 2010) have been demonstrated to increase under elevated $\mathrm{CO_2}$ concentration, contrary to the findings during this experiment (Hornick, 2015). All of these responses are to a large extent dependent on the plankton community composition. For example, the increased bacterial production observed in a mesocosm study
in a Norwegian fjord was probably a response to increased carbon availability produced by phytoplankton (Grossart et al., 2006). DOC production by phytoplankton is determined by the physiological state and the composition of the community (Thornton, 2014); in particular diatoms have been intensively studied in this respect and are known to be important DOC producers (Hoagland et al., 1993). Shifts in the phytoplank-
ton community may alter the DOC production (Spilling et al., 2014), and any shifts in the plankton community composition, caused by ocean acidification, may have greater effects on ecosystem functioning than any direct effect of increasing $f\mathrm{CO_2}$/decreasing pH (Eggers et al., 2014).

**[BGD]**

doi:10.5194/bg-2015-608

**Ocean acidification decreases respiration**

K. Spilling et al.

It is evident that there were other parameters that influence the physiology of the plankton community as a whole outside the mesocosms. Changes in community composition and nutrient availability seem the most plausible reasons. A better understanding of how different physical, chemical and biological factors interact with each other is needed in order to improve our understanding of how marine ecosystems change under the influence of a range of environmental pressures.

## 4.5 Potential implications for carbon cycling

A lot of attention during past decades has been directed to understanding the biological carbon pump, as it is a key mechanism for sequestering atmospheric $CO_2$. The potential export is ultimately determined by gross primary production minus total community respiration. Even small changes in the production or loss term of this equation have the potential to greatly affect biogeochemical cycling of carbon.

The exported carbon decreased during the experiment. Part of this decrease was probably due to sinking of existing organic material at the start of the experiment and can be seen as the reduction in TPC. However, this also coincided with the shift towards increased dominance of picoplankton. Size is a key parameter determining sinking speed, and picoplankton is very inefficient in transporting carbon out of the euphotic layer (Michaels and Silver, 1988). The shift towards smaller size classes was likely also contributing to the reduction in exported carbon.

The 40 % reduction in respiration with increasing $f\mathrm{CO}_2$ found in our study could have great implications for net export of carbon in the future ocean. There is, however, uncertainty in the results, in particular that the measured net carbon fixation under increased $CO_2$ was not higher than in the controls. In the case of reduced respiration, an increase in net primary production can be expected, as loss rates are reduced. That the measured carbon fixation was not evidently different between treatments could be due to similar reduction in GPP, as indicated by carbon flux estimates (Spilling et al. 2015). Alternatively, the measurement uncertainty in our small scale incubations (8 mL), involving several pipetting steps, was likely higher than the respiration measurements,

**BGD**

doi:10.5194/bg-2015-608

**Ocean acidification decreases respiration**

K. Spilling et al.

which could have prevented us from picking up any $CO_2$ effect on primary production. Another complicating factor is what the $^{14}C$ method is actually measuring (Sakshaug et al., 1997; Falkowski and Raven, 2013). The consensus seems to be somewhere between gross and net production, but leaning towards net production with long incu-
bation times (Marra, 2009).

There was evidence of a positive $CO_2$ effect on the amount of Chl *a*, TPC and DOC pools (Paul et al., 2015), suggesting that the reduced respiration does translate into higher net carbon fixation. This effect was seen from the latter part of Phase II and the trend continued after t31 (these parameters were sampled until t43). This increased
net carbon fixation did not, however, affect carbon export as there was no detectable difference in the sinking flux measurements (Table 1 and Paul et al., 2015). The results suggests that the increased carbon fixation ended up in the smallest size fraction of TPC not being exported and/or into the dissolved organic carbon pool. Further support for this conclusion is presented in Paul et al. (2015), Crawfurd et al. (2015) and
Lischka et al. (2015). Under this scenario, increased carbon fixation in a high $CO_2$ environment will not be a negative feedback mechanism for increasing atmospheric $CO_2$ concentration.

*Acknowledgements.* We would like to thank all of the staff at Tvärminne Zoological station for great help during this experiment, and Michael Sswat for carrying out the TPC filtrations.
We also gratefully acknowledge the captain and crew of R/V ALKOR (AL394 and AL397) for their work transporting, deploying and recovering the mesocosms. The collaborative mesocosm campaign was funded by BMBF projects BIOACID II (FKZ 03F06550) and SOPRAN Phase II (FKZ 03F0611). Additional financial support for this study came from Academy of Finland (KS - Decisions no: 259164 and 263862) and Walter and Andrée de Nottbeck Foundation (KS, NV).

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

**Table 1.** Average net primary production (NPP), total respiration (TR), exported total particulate carbon (Exp TPC) and change in TPC (ΔTPC) in mmol C m$^{-2}$ d$^{-1}$ during Phase I and Phase II of the experiment. TPC standing stock is the average of 3 sampling dates at the beginning of the period (t3–t1 and t15–t19) in mmol C m$^{-2}$ ± SE. NPP and TR was corrected for the missing measuring days during Phase I. TR was measured as $O_2$ consumption and for comparison with carbon fixation we used a respiratory quotient (RQ) of 1.

| Phase I (t0–t16) | | | | | | |
|---|---|---|---|---|---|---|
| $CO_2$ treatment (µatm $f$$CO_2$) | 346 | 348 | 494 | 868 | 1075 | 1333 |
| NPP | 4.8 | 11.4 | 14.9 | 12.3 | 11.3 | 14.5 |
| TR | 107 | 82 | 81 | 80 | 75 | 74 |
| Exp TPC | 7.4 | 6.3 | 6.1 | 6.8 | 6.4 | 6.9 |
| ΔTPC | −6.9 | −5.0 | −7.0 | −6.7 | −6.8 | −6.9 |
| TPC | 442 ± 38 | 408 ± 39 | 435 ± 48 | 426 ± 38 | 418 ± 48 | 446 ± 57 |
| **Phase II (t17–t31)** | | | | | | |
| $CO_2$ treatment (µatm $f$$CO_2$) | 346 | 348 | 494 | 868 | 1075 | 1333 |
| NPP | 3.8 | 11.2 | 10.8 | 14.3 | 10.4 | 12.0 |
| TR | 140 | 127 | 103 | 103 | 101 | 86 |
| Exp TPC | 3.3 | 2.6 | 2.5 | 2.6 | 2.8 | 2.9 |
| ΔTPC | −2.1 | −2.1 | −1.9 | 0.8 | 2.9 | 2.9 |
| TPC | 325 ± 14 | 323 ± 20 | 316 ± 20 | 313 ± 9 | 302 ± 12 | 328 ± 23 |

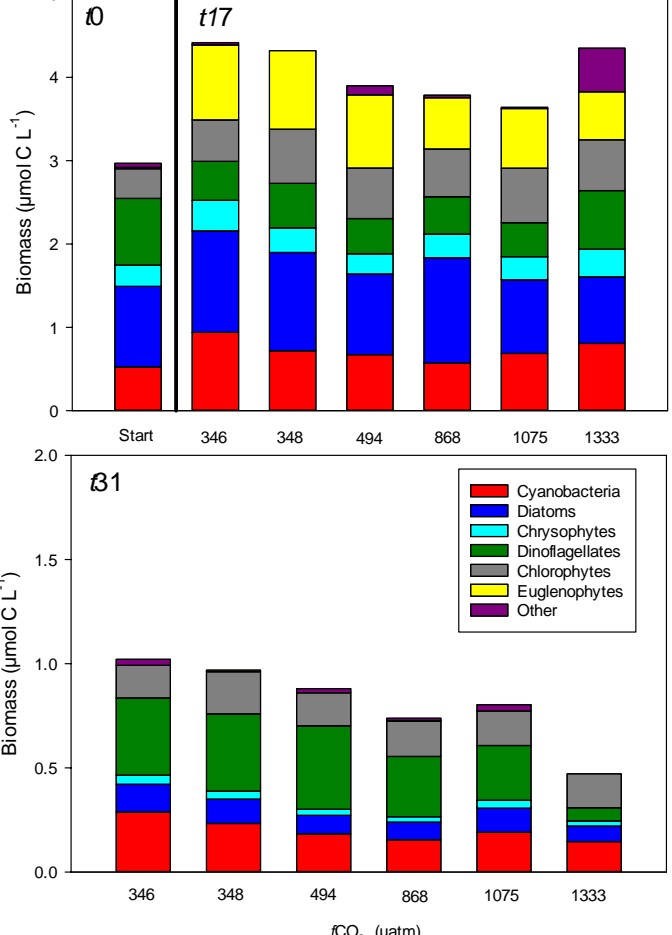

**Figure 1.** The main phytoplankton groups at the start of the experiment, t0, and t17 (upper panel) and t31 (lower panel). The initial (t0) was the average of all mesocosm bags.

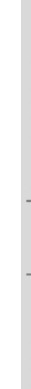

**BGD**

doi:10.5194/bg-2015-608

**Ocean acidification decreases respiration**

K. Spilling et al.

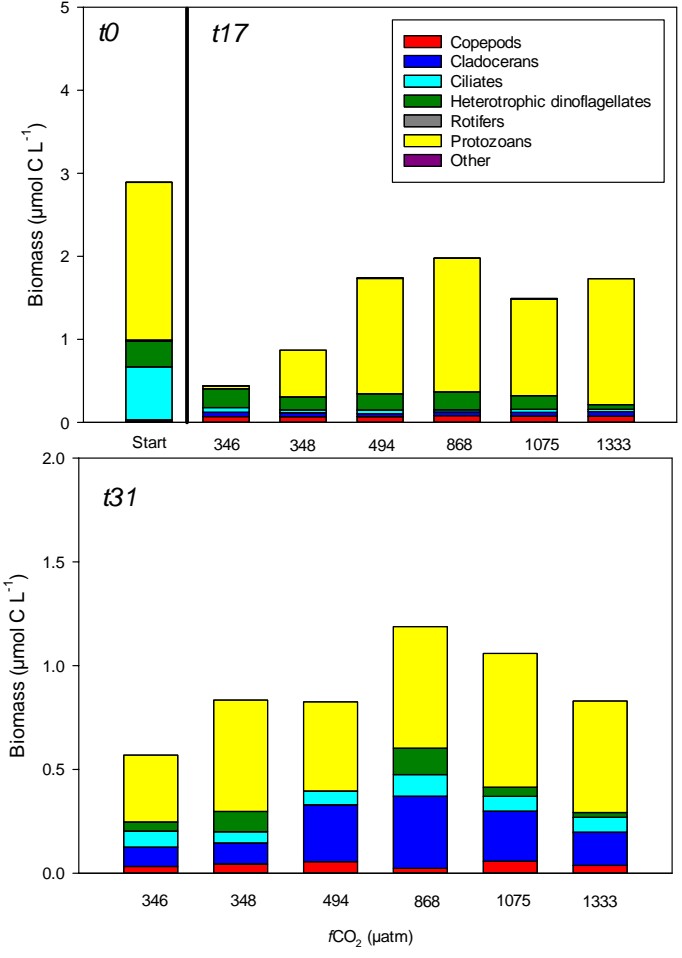



Discussion Paper | Discussion Paper | Discussion Paper | Discussion Paper

**BGD**

doi:10.5194/bg-2015-608

**Ocean acidification decreases respiration**

K. Spilling et al.

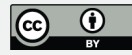

**Figure 2.** The main micro- and mesozooplankton groups at the start of the experiment, t0, and t17 (upper panel) and t31 (lower panel). The initial (t0) was the average of all mesocosm bags.

Discussion Paper | Discussion Paper | Discussion Paper | Discussion Paper

**BGD**

doi:10.5194/bg-2015-608

**Ocean acidification decreases respiration**

K. Spilling et al.



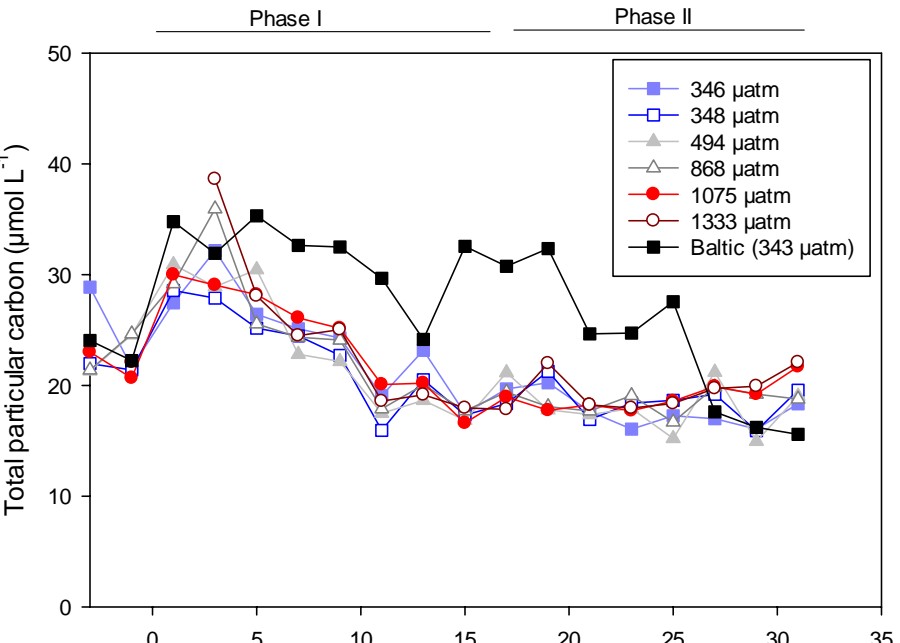

**Figure 3.** The development of total particulate carbon (TPC) during the experiment.

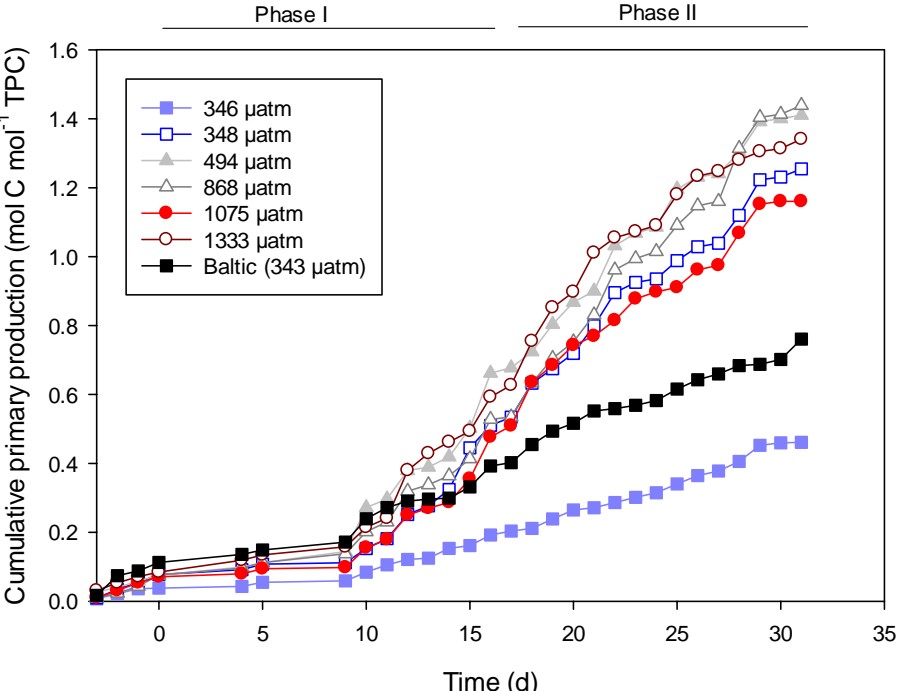

**Figure 4.** The cumulative primary production in the different $f\mathrm{CO_2}$ treatments normalized to total particulate carbon (TPC) in the euphotic zone. The $f\mathrm{CO_2}$ (µatm) were the average measured over the duration of the experiment. The two lowest $f\mathrm{CO_2}$ treatments (346 and 348 µatm) were controls without any $\mathrm{CO_2}$ addition. The two phases of the experiment is indicated by the horizontal bars on top.

Discussion Paper | Discussion Paper | Discussion Paper | Discussion Paper | Discussion Paper |

BGD

doi:10.5194/bg-2015-608

**Ocean acidification decreases respiration**

K. Spilling et al.

Discussion Paper | Discussion Paper | Discussion Paper | Discussion Paper | Discussion Paper |

**BGD**

doi:10.5194/bg-2015-608

**Ocean acidification decreases respiration**

K. Spilling et al.

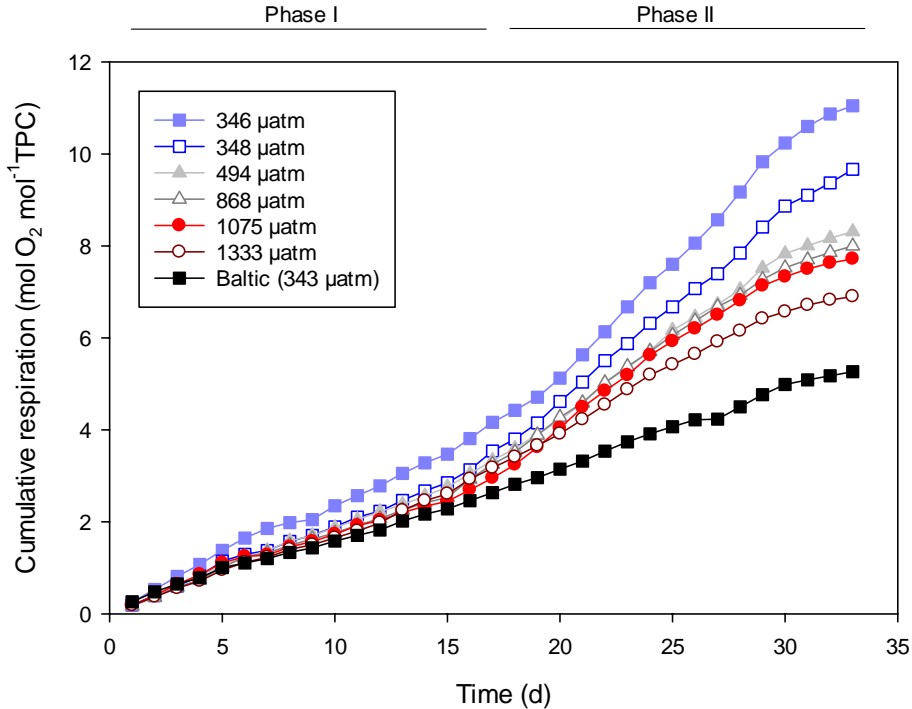

**Figure 5.** The cumulative respiration in the different $f\mathrm{CO}_2$ treatments normalized to total particulate carbon (TPC). The $f\mathrm{CO}_2$ (µatm) were the average measured over the duration of the experiment. The two lowest $f\mathrm{CO}_2$ treatments (346 and 348 µatm) were controls without any $\mathrm{CO}_2$ addition. The two phases of the experiment is indicated by the horizontal bars on top.

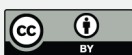

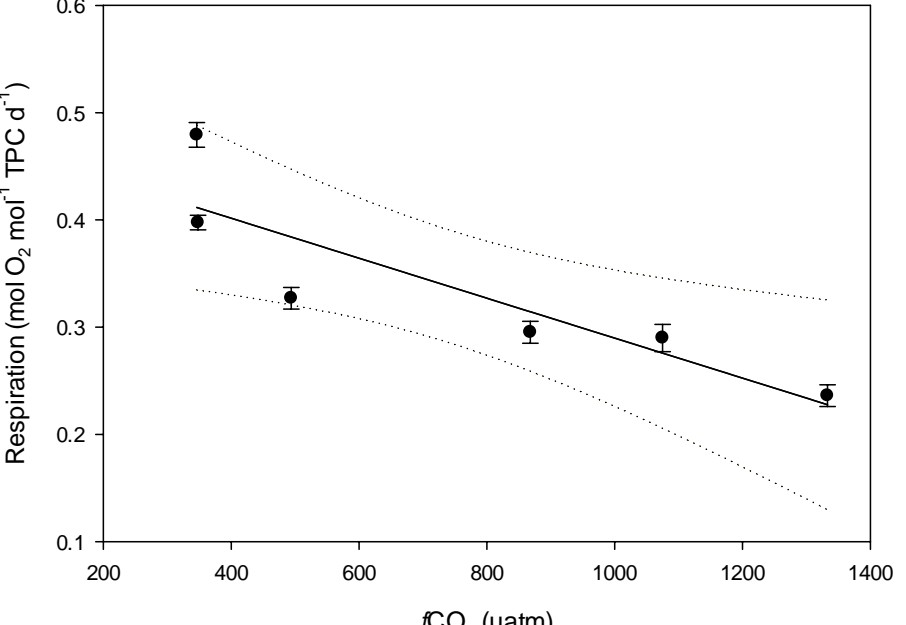

**Figure 6.** The respiration rate, normalized to total particulate carbon (TPC), in the different $fCO_2$ treatments during the latter half of the experiment (t20–t31). Respiration was estimated by linear regression from the data presented in Fig. 4 from the time when an effect of increased $CO_2$ concentration was first observed. The error bars represents standard error (SE) of the residuals from the linear regression. The solid line represents the linear regression (slope −0.0002; $p = 0.02$; $R^2 = 0.77$) and dotted lines the 95 % confidence intervals.

Discussion Paper | Discussion Paper | Discussion Paper | Discussion Paper |

**BGD**

doi:10.5194/bg-2015-608

**Ocean acidification decreases respiration**

K. Spilling et al.