# Peer review of "2 Ocean acidification decreases plankton respiration: evidence from a mesocosm experiment 4 5 6 K. Spilling1, 2, A. J. Paul3 N. Virkkala2, T. Hastings4, S. Lischka3 A. Stuhr3, R. Bermudez3, 5, 7 J. Czerny3, T"

_Biogeosciences, 2015_

## Referee Comment (RC1) · P. Neale (Referee) · 25 Feb 2016

The authors report on some results of a whole ecosystem CO2 enrichment study using large volume mesocosms moored in the Gulf of Finland as part of a special issue on "Effects of rising CO2 on a Baltic Sea plankton community: ecological and biogeochemical impacts". This report makes a substantial contribution to the issue by focusing on the important processes of carbon exchange through primary production and respiration. Overall the study is complex and multidimensional, and it would have been difficult to review this article outside the context of a special issue as many important details of the experimental design and basic observations on the mesocosms appear in other reports. Fortunately, with the open discussion format of Biogeosciences Discussions, these other reports were accessible to the reviewer. A key result is that respiration decreased as a function of CO2 enrichment though the difference only emerge

towards the end of the experiment when composition of the mesocosms (even controls) had substantially diverged from surrounding waters. But the question is left open as to what happened to the extra carbon, as the study did not observe a concomitant increase in net primary production.

While the results do demonstrate decreased respiration for samples from the higher $CO_2$ enrichments, I do have some concern about how representative these rates are of processes in the mesocosms. A depth integrated water sample was taken and incubated at "ambient" temperature. But it can be seen from Paul et al (2015) that there was a strong temperature gradient over the mesocosm's depth range, at times as much as $10°C$, so it is not clear what was "ambient" temperature. Moreover, mixing waters of differing temperatures may bias the respiration measurement at a fixed temperature vs. the "real" average, i.e. combining warm, lower particle concentration surface water with cooler, high particle (or nutrient) concentration bottom water could stimulate respiration versus the average of the two.

The authors also indicate that respired carbon was about 10x greater than net production (pg. 17 line 7). Some more explanation is needed for why such comparison is made since a determination of whether the system is net heterotrophic or autotrophic would require comparison of gross primary production with total community respiration, as stated on page 21 line 9. The statement on page 21 line 26 implies that the authors have some idea of gross primary production, could this be compared to respiration rate?

The authors also speculate that the net primary productivity method may not have been sensitive enough to detect difference between treatments, so that enhanced production at increased $CO_2$ was not detected. Small incubation volumes are suggested to contribute to uncertainty but the authors give no indication of what was that measurement uncertainty. Nevertheless, they state that the measurements were comparable to previous ones in the same regions using similar methods (Kivi et al. 1993) which would argue against any substantial bias. One other factor to consider as to whether

the NPP assay would detect an enhancement effect was that the incubations were conducted outside the bags. According to Riebesell et al. (2013), the mesocosm material (thermoplastic polyurethane) removes all UV whereas glass scintillation vials used for the NPP incubation transmit UV-A and most UV-B so rates in the vials could have been substantially more inhibited in the near surface samples than phytoplankton in the mesocosms that were protected from UV. Moreover, some studies have shown that phytoplankton grown under $CO_2$ enhanced conditions are more sensitive to UV. It is possible that NPP was higher in the mesocosms with $CO_2$ enrichment but the effect was dampened in incubations outside the bag due to a counterbalancing increase in sensitivity to UV (see, e.g., Sobrino et al. 2008, 2009). Also, as the lead author knows (since he was co-author on the paper), Sobrino et al. (2014) observed lower rates of DOC release during short term PPR incubations by phytoplankton acclimated to $CO_2$ enhanced conditions but this effect was much less when incubations included UV. This DOC would be quite labile and rapidly respired so might not affect the bulk DOC pool but a reduction in DOC release could decrease bacterial respiration.

Specific Comments:

The lack of UV in the bags should be mentioned in the text, e.g. :

Pg. 20 line 5 " light and temperature were similar both inside and outside the mesocosm bags. "

Except that UV was absent inside the mesocosms.

Page 22 – The discussion finishes abruptly, a summary paragraph would be helpful

Respectfully submitted,

Patrick Neale Edgewater, MD, USA

Sobrino, C., Neale, P.J., Phillips-Kress, J.D., Moeller, R.E., and Porter, J. (2009) Elevated $CO_2$ increases sensitivity to ultraviolet radiation in lacustrine phytoplankton assemblages. Limnol. Oceanogr. 54: 2448-2459.

Sobrino, C., Ward, M.L., and Neale, P.J. (2008) Acclimation to elevated carbon dioxide and ultraviolet radiation in the diatom Thalassiosira pseudonana: Effects on growth, photosynthesis, and spectral sensitivity of photoinhibition. Limnol. Oceanogr. 53: 494-505.

Sobrino, C., Segovia, M., Neale, P.J., Mercado, J.M., García-Gomez, C., Kulk, G., Lorenzo, M.R., Camarena, T., van de Poll, W.H., Spilling, K. and others. (2014) Effect of CO2, nutrients and light on coastal plankton. IV. Physiological responses. Aquatic Biology 22: 77-93.

———————————————————————

---

## Referee Comment (RC2) · Anonymous Referee #2 · 9 Mar 2016

GENERAL COMMENTS: The manuscript by Spilling et al. reports on the response of the plankton community to a gradient of increasing CO2 concentrations, focusing on the effects that this treatments had on respiration, carbon fixation and carbon export. Authors conclude that respiration decreases under high CO2 concentrations, while primary productivity did not increase as a consequence of such CO2 levels (contrary to the already observed in many studies carried out up to date). The aim of this study was to provide new knowledge of the effects of elevated CO2 in a system such as the Baltic, were no many data sets on this topic are recorded. Therefore the work hereby presented focuses on a relevant and timely topic for scientists working on the effect of global change on aquatic ecosystems.

However, I have several concerns that (in my opinion) warrant further attention from the authors. I found surprising the lack of real independent mesocosms replicates. Only

the controls do replicate (M1 and M5). Under these circumstances an appropriate statistical analysis cannot be performed, compromising the significance of the results. In its place, regression coefficient significance tests have been done to analyse the significance of the results. Although valid, these tests compare the mesocosms between them, but the behaviour, obviously implying variability within each specific treatment cannot then be ruled out, because without replicates is not possible to discern if the response is due to the controlled factor ($CO_2$) or to any other uncontrolled factor, and or their interaction. At least, significance differences obtained from the R comparisons tests should be mentioned in the text adding the p values (in results section) and marked in the Figures as an asterisk or letter to indeed demonstrate that there are some differences. A table including the results of all linear regression analyses indicating the significant effects of the different $CO_2$ concentrations on the variables would needed (see Tables and Figures in Paul et al. 2015, Crawfurd et al., 2015, Bermudez et al., 2015-this special issue-as examples of what I am referring to). In my opinion, in this manner the Ms would benefit of a better understanding of the results.

The other important issue is that you mention measurement uncertainties at some points. I do not understand how or why can be a measurement uncertainty working with small volumes, can you specify? How this affect reliability? Regarding the incubation time with 14C, I think it is widely demonstrated that this method is quite sensitive. I agree that it may be more estimative of NPP, but, in incubations long as 24h, the same 14C molecule can be fixed and respired several times (the eternal discussion). Do you think you could be getting an underestimate of your measurement? Said this, I think the point raised by reviewer 1 regarding the effect of UV on C fixation during incubations would be much more relevant in terms of affecting PP (not commenting on UVR as I totally agree and support reviewer 1 comments). Also said by reviewer 1, if you think there are uncertainties, how your data compare to former published studies?

It is not clear to me whether you also mention measurement uncertainties on the TPC data, it seems so. In this regards, if there exist such an uncertainty in TPC, how

this translates into figures 4 and 5 that are normalised by TPC? The -under or -sub estimations would then be included in your calculations on the cumulative PP and TR and vertical C flux?

SPECIFIC COMMENTS: Pg. 12.Phytoplankton community composition. As data are presented it is not clearly seen that there is dominance of some groups over others. Only Euglenophytes seem to be absent in t0. Dinoflagellates, cyanobacteria, diatoms and chlorophytes look like having similar proportions in t0 and t17 (p values needed), while "other" increase at 1333 ppm. What organisms does "other" comprise? Stacked area plots would give a much better idea of the temporal evolution and trend followed by the community and so significances could be better appreciated. Thus I suggest to re-plot figures 1 and 2 including all days and treatments in stacked area charts. How your data compare to Bermudez et al. this issue-seems that taxonomy differs a little in between the two studies (for instance Euglenophyta).

Considering that your study deals with the plankton food web, bacterial production, or at least abundances have not been analysed. Although probably low in volume and biomass contribution as compared to phyto and zooplankton groups, they are important too since they have been reported to react positively to increased CO2 (a number of papers published on this topic by Grossart, Schulz and Riebesell). I see bacterial contribution is further discussed in pg. 20 based on former reports. How about bacterial production/abundances in this very mesocosms experiment? Neither you say anything about viruses affecting C losses, which is important for C cycling and definitively affect C export. These two (bacteria and viruses) in my opinion shall be at least being discussed (succinctly if you wish) within the framework of the whole mesocosm experiment.

Pg. 18. Ln 20. "The larger-scale mesocosms...interacting effects between different components of the food web are included". Pg. 19 Ln 21. Subheading "Interacting effects and community composition". Also in pg. 20 Ln 10 interactive effects are mentioned. I find this an overstatement since you have not analysed interactive effects

statistically, thus you cannot conclude anything on that.

Pg. 18 Ln 22-pg. 19 Ln 6. Instead of discussing higher plants which do not deal with carbonic /carbonate equilibrium and the systems are different, I think it would make much more sense to focus on explaining the mechanisms why respiration might be reduced in aquatic organisms such as phytoplankton at high CO2. Can the decreased TR be related to CCMs? Both photosynthesis and respiration generate energy that can be used for CCMs since they are mechanisms highly-energy -demanding. Under increased CO2 it is well known that CCMs are downregulated. If there are no active CCMs, then respiration and photosynthesis might also be downregulated, and the energy consumed by them is "available" for other purposes. On the other hand, such energy could also be directed to growth (i.e. PP) that is what you are describing. This would mean that respiration could be downregulated but not PP. Such uncoupling is what is important to discuss in depth. Also, how is this related to pigment concentration? Since under high CO2 there is less electronic demand, pigments should decrease. Indeed Chla sharply decreased from 2 ugL-1 on P1 to 0.8 in PII and III (Paul et al., 2015). However you estate in pg. 22 Ln 6 that CO2 had a positive effect on Chla. Some clarification is needed.

The paragraph on the effect of pH is interesting but maybe worthwhile looking at more updated papers on pH microenvironment in phytoplankton (Flynn et al., 2012. NATURE CLIMATE CHANGE | VOL 2 | JULY 2012 ; Taylor et al., 2012, Trends in Plant Science, November 2012, Vol. 17, No. 11 )

General: I think the discussion need more work and a better connexion between sections. Agree with reviewer 1 on addition of a summary paragraph at the end.

TECHNICAL COMMENTS: Substitute "parameter" by "variable". What you are measuring are variables. Parameters are constants that relate variables.

---

## Author Comment (AC1) · 9 Apr 2016

Response to review

We are grateful for all the constructive comments and suggestions, which have improved the manuscript. Below we have replied to all of the issues raised by the reviewers.

Dr. Neale Reviewer #1, comment #1. While the results do demonstrate decreased respiration for samples from the higher $CO_2$ enrichments, I do have some concern about how representative these rates are of processes in the mesocosms. A depth integrated water sample was taken and incubated at "ambient" temperature. But it can be seen from Paul et al (2015) that there was a strong temperature gradient over the mesocosm's depth range, at times as much as 10_C, so it is not clear what was

"ambient" temperature. Moreover, mixing waters of differing temperatures may bias the respiration measurement at a fixed temperature vs. the "real" average, i.e. combining warm, lower particle concentration surface water with cooler, high particle (or nutrient) concentration bottom water could stimulate respiration versus the average of the two.

Author response: It is true that temperature stratification varied. We kept the incubation temperature at the surface temperature, and we will add this information to the Materials and Methods chapter. Dr. Neale makes a valid point that there might be a bias due to mixing water of different temperature rather than averaging measurements taken at different temperature. Logistical constraints prevented us from making respiration incubations at several temperatures. We will take up this potential bias in the Discussion chapter.

Reviewer #1, comment #2. The authors also indicate that respired carbon was about 10x greater than net production (pg. 17 line 7). Some more explanation is needed for why such comparison is made since a determination of whether the system is net heterotrophic or autotrophic would require comparison of gross primary production with total community respiration, as stated on page 21 line 9. The statement on page 21 line 26 implies that the authors have some idea of gross primary production, could this be compared to respiration rate?

Author response: We did try to estimate the gross primary production and after the submission of this paper we made a carbon budget for the whole experiment, which is presented in a synthesis paper (Spilling et al 2016). We will remove these statements from the present paper and place a reference to the budget paper in order to have a more clear focus.

Reviewer #1, comment #3. The authors also speculate that the net primary productivity method may not have been sensitive enough to detect difference between treatments, so that enhanced production at increased $CO_2$ was not detected. Small incubation volumes are suggested to contribute to uncertainty but the authors give no indication

of what was that measurement uncertainty. Nevertheless, they state that the measurements were comparable to previous ones in the same regions using similar methods (Kivi et al. 1993) which would argue against any substantial bias. One other factor to consider as to whether the NPP assay would detect an enhancement effect was that the incubations were conducted outside the bags. According to Riebesell et al. (2013), the mesocosm material (thermoplastic polyurethane) removes all UV whereas glass scintillation vials used for the NPP incubation transmit UV-A and most UV-B so rates in the vials could have been substantially more inhibited in the near surface samples than phytoplankton in the mesocosms that were protected from UV. Moreover, some studies have shown that phytoplankton grown under $CO_2$ enhanced conditions are more sensitive to UV. It is possible that NPP was higher in the mesocosms with $CO_2$ enrichment but the effect was dampened in incubations outside the bag due to a counterbalancing increase in sensitivity to UV (see, e.g., Sobrino et al. 2008, 2009). Also, as the lead author knows (since he was co-author on the paper), Sobrino et al. (2014) observed lower rates of DOC release during short term PPR incubations by phytoplankton acclimated to $CO_2$ enhanced conditions but this effect was much less when incubations included UV. This DOC would be quite labile and rapidly respired so might not affect the bulk DOC pool but a reduction in DOC release could decrease bacterial respiration.

Author response: A very good point that we will take up in the Discussion chapter. The DOC concentration in the Baltic Sea is very high compared with most other oceans and coastal seas (like the Mediterranean that is referred to). Most of this is refractory DOC, which effectively absorbs in the UV region, and typically the depth at which 1% of UVB remains is <50 cm (e.g. Piazena and Häder 1994). UVA penetrates a little deeper and may have affected slightly the incubation platform moored at 2 m depth. We do not believe, however, that UV light have caused major inhibition of our primary production measurements (or affected labile DOC production), but we will point this out with the reasoning described above.

We will make appropriate changes to all the specific comments.

Reviewer #2

Reviewer #2, comment #1. I have several concerns that (in my opinion) warrant further attention from the authors. I found surprising the lack of real independent mesocosms replicates. Only the controls do replicate (M1 and M5). Under these circumstances an appropriate statistical analysis cannot be performed, compromising the significance of the results. In its place, regression coefficient significance tests have been done to analyse the significance of the results. Although valid, these tests compare the mesocosms between them, but the behaviour, obviously implying variability within each specific treatment cannot then be ruled out, because without replicates is not possible to discern if the response is due to the controlled factor ($CO_2$) or to any other uncontrolled factor, and or their interaction. At least, significance differences obtained from the R comparisons tests should be mentioned in the text adding the p values (in results section) and marked in the Figures as an asterisk or letter to indeed demonstrate that there are some differences. A table including the results of all linear regression analyses indicating the significant effects of the different $CO_2$ concentrations on the variables would needed (see Tables and Figures in Paul et al. 2015, Crawfurd et al., 2015, Bermudez et al., 2015-this special issue-as examples of what I am referring to). In my opinion, in this manner the Ms would benefit of a better understanding of the results.

Author response: The mesocosm bags are relatively large scale operations, 55 m3 in each enclosure, and this puts some constrains on how many units can be used. Lack of replication does not, however, prevent proper statistical analysis of results: for example gradient experiments of a single variable or factorial design experiments with multiple variables, both provide data that can be statistically tested for treatment effects (see e.g. the discussion by Oksanen 2001 and Hurlbert 2004). In our case, a gradient of different $CO_2$ additions was used. The statistical test was in the figure legend, and we will include it also in the results section as suggested by the reviewer.

Reviewer #2, comment #2. The other important issue is that you mention measurement uncertainties at some points. I do not understand how or why can be a measurement uncertainty working with small volumes, can you specify? How this affect reliability? Regarding the incubation time with 14C, I think it is widely demonstrated that this method is quite sensitive. I agree that it may be more estimative of NPP, but, in incubations long as 24h, the same 14C molecule can be fixed and respired several times (the eternal discussion). Do you think you could be getting an underestimate of your measurement? Said this, I think the point raised by reviewer 1 regarding the effect of UV on C fixation during incubations would be much more relevant in terms of affecting PP (not commenting on UVR as I totally agree and support reviewer 1 comments). Also said by reviewer 1, if you think there are uncertainties, how your data compare to former published studies?

Author response: There will always be measurements uncertainties (depending on the methodology, instrument etc) and this would be independent of the volume, and we are not quite sure that we got your point. Perhaps you refer to the primary production measurements. In that case the incubation volume was relatively small, and we did not remove the grazers, which could have introduced a bias with respect to grazing pressure, i.e. the number of grazers could have been quite variable depending on how many by chance got into the relatively small incubation volume. With respect to the UV point, please see our response above to reviewer #1.

Reviewer #2, comment #3. It is not clear to me whether you also mention measurement uncertainties on the TPC data, it seems so. In this regards, if there exist such an uncertainty in TPC, how this translates into figures 4 and 5 that are normalised by TPC? The -under or –sub estimations would then be included in your calculations on the cumulative PP and TR and vertical C flux?

Author response: There are of course also measurements uncertainties in the TPC, and yes these would be included in the data presented in Figs 4 and 5. However, it does not affect the main conclusion of the paper.

Reviewer #2, comment #4. Phytoplankton community composition. As data are presented it is not clearly seen that there is dominance of some groups over others. Only Euglenophytes seem to be absent in t0. Dinoflagellates, cyanobacteria, diatoms and chlorophytes look like having similar proportions in t0 and t17 (p values needed), while "other" increase at 1333 ppm. What organisms does "other" comprise? Stacked area plots would give a much better idea of the temporal evolution and trend followed by the community and so significances could be better appreciated. Thus I suggest to re-plot figures 1 and 2 including all days and treatments in stacked area charts. How your data compare to Bermudez et al. this issue-seems that taxonomy differs a little in between the two studies (for instance Euglenophyta).

Author response: We wanted to present a general overview of the plankton community composition, and a more in-depth analysis and presentation of all the dates are provided in Bermudez et al 2016 and Lischka et al 2015. The presented phytoplankton data is the same as Bermudez et al 2016, but here we have additionally included counts of phytoplankton >20$\mu$m, affecting the biomass of e.g. Euglenophyta.

Reviewer #2, comment #5. Considering that your study deals with the plankton food web, bacterial production, or at least abundances have not been analysed. Although probably low in volume and biomass contribution as compared to phyto and zooplankton groups, they are important too since they have been reported to react positively to increased CO2 (a number of papers published on this topic by Grossart, Schulz and Riebesell). I see bacterial contribution is further discussed in pg. 20 based on former reports. How about bacterial production/abundances in this very mesocosms experiment? Neither you say anything about viruses affecting C losses, which is important for C cycling and definitively affect C export. These two (bacteria and viruses) in my opinion shall be at least being discussed (succinctly if you wish) within the framework of the whole mesocosm experiment.

Author response: A very good point and we will incorporate this into the discussion. Bacterial production was measured (Hornick et al 2016), but this was not out yet at the

time of review. Heterotrophic prokaryotes were enumerated and this data is presented in Crawfurd et al (2016).

Reviewer #2, comment #6. Pg. 18. Ln 20. "The larger-scale mesocosms . . . inter-acting effects between different components of the food web are included". Pg. 19 Ln 21. Subheading "Interacting effects and community composition". Also in pg. 20 Ln 10 interactive effects are mentioned. I find this an overstatement since you have not analysed interactive effects

Author response: This section is under the discussion of advantages of mesocosm experiments on a general level. We will change this to "possible interacting effects. . ." to make it more clear.

Reviewer #2, comment #7. Pg. 18 Ln 22-pg. 19 Ln 6. Instead of discussing higher plants which do not deal with carbonic /carbonate equilibrium and the systems are dif-ferent, I think it would make much more sense to focus on explaining the mechanisms why respiration might be reduced in aquatic organisms such as phytoplankton at high $CO_2$. Can the decreased TR be related to CCMs? Both photosynthesis and respiration generate energy that can be used for CCMs since they are mechanisms highly-energy -demanding. Under increased $CO_2$ it is well known that CCMs are downregulated. If there are no active CCMs, then respiration and photosynthesis might also be down-regulated, and the energy consumed by them is "available" for other purposes. On the other hand, such energy could also be directed to growth (i.e. PP) that is what you are describing. This would mean that respiration could be downregulated but not PP. Such uncoupling is what is important to discuss in depth. Also, how is this related to pig-ment concentration? Since under high $CO_2$ there is less electronic demand, pigments should decrease. Indeed Chla sharply decreased from 2 ugL-1 on P1 to 0.8 in PII and III (Paul et al., 2015). However you estate in pg. 22 Ln 6 that $CO_2$ had a positive effect on Chla. Some clarification is needed.

Author response: A very good point and a more thorough discussion around CCMs will

be incorporated into the Discussion chapter. As for the Chl a, the major change was driven by change in total phytoplankton biomass, e.g. the overall decrease from PI to PII and PIII, and the higher Chl a in the high $CO_2$.

We will make appropriate changes to all the specific comments and technical suggestions raised by the reviewer.

References

Bermúdez, J. R., Winder, M., Stuhr, A., Almén, A. K., Engström-Öst, J., and Riebesell, U.: Effect of ocean acidification on the structure and fatty acid composition of a natural plankton community in the Baltic Sea, Biogeosciences Discuss., doi:10.5194/bg-2015-669, in review, 2016.

Crawfurd, K. J., Brussaard, C. P. D., and Riebesell, U.: Shifts in the microbial community in the Baltic Sea with increasing $CO_2$, Biogeosciences Discuss., doi:10.5194/bg-2015-606, in review, 2016.

Hornick, T., Bach, L. T., Crawfurd, K. J., Spilling, K., Achterberg, E. P., Brussaard, C. P. D., Riebesell, U., and Grossart, H.-P.: Ocean acidification indirectly alters trophic interaction of heterotrophic bacteria at low nutrient conditions, Biogeosciences Discuss., doi:10.5194/bg-2016-61, in review, 2016.

Hurlbert, S.: On misinterpretation of pseudoreplication and related matters: a reply to Oksanen. Oikos 104: 591-597, 2004.

Lischka, S., Bach, L. T., Schulz, K.-G., and Riebesell, U.: Micro- and mesozooplankton community response to increasing $CO_2$ levels in the Baltic Sea: insights from a large-scale mesocosm experiment, Biogeosciences Discuss., 12, 20025-20070, doi:10.5194/bgd-12-20025-2015, 2015.

Oksanen, L.: Logic of experiments in ecology: is pseudoreplication a pseudoissue? Oikos 94: 27-38, 2001.

Paul, A. J., Bach, L. T., Schulz, K.-G., Boxhammer, T., Czerny, J., Achterberg, E. P., Hellemann, D., Trense, Y., Nausch, M., Sswat, M., and Riebesell, U.: Effect of elevated $CO_2$ on organic matter pools and fluxes in a summer Baltic Sea plankton community, Biogeosciences, 12, 6181-6203, doi:10.5194/bg-12-6181-2015, 2015.

Piazena, H., Häder, D-P.: Penetration of solar UV irradiation in coastal lagoons of the southern Baltic Sea and its effect on phytoplankton communities. Photochemistry and Photobiology 60: 463-469, 1994.

Sobrino, C., Neale, P.J., Phillips-Kress, J.D., Moeller, R.E., and Porter, J.: Elevated $CO_2$ increases sensitivity to ultraviolet radiation in lacustrine phytoplankton assemblages. Limnol. Oceanogr. 54: 2448-2459, 2009.

Sobrino, C., Ward, M.L., and Neale, P.J.: Acclimation to elevated carbon dioxide and ultraviolet radiation in the diatom Thalassiosira pseudonana: Effects on growth, photosynthesis, and spectral sensitivity of photoinhibition. Limnol. Oceanogr. 53: 494- 505, 2008.

Sobrino, C., Segovia, M., Neale, P.J., Mercado, J.M., García-Gomez, C., Kulk, G., Lorenzo, M.R., Camarena, T., van de Poll, W.H., Spilling, K. and others.: Effect of $CO_2$, nutrients and light on coastal plankton. IV. Physiological responses. Aquatic Biology 22: 77-93, 2014.

Spilling, K., Schulz, K. G., Paul, A. J., Boxhammer, T., Achterberg, E. P., Hornick, T., Lischka, S., Stuhr, A., Bermúdez, R., Czerny, J., Crawfurd, K., Brussaard, C. P. D., Grossart, H.-P., and Riebesell, U.: Effects of ocean acidification on pelagic carbon fluxes in a mesocosm experiment, Biogeosciences Discuss., doi:10.5194/bg-2016-56, in review, 2016.

---

## Author Response (AR1)

Dear Editor

Please find attached a revised manuscript where we have addressed all the points made by the two reviewers. In the first part we have gone through all the points that you have made, and indicated changes to the revised version of the manuscript highlighted in red. In the second part we have added our original response to the review and added the concrete changes made in the revised manuscript.

In your decision statement we believe you have mixed the reviewers' #, as reviewer #1 had only 3 general comments, i.e. comments 4 and 5 that you refer to must be from reviewer #2 (which also fits with the general comments). Below we have kept the original numbering of the reviewers.

**Reply to the Editor's comments**

**Editor: Reviewer 1, comment 2**
Please provide better insight; state values used and methods used

**Author response:** The reviewer's comments refer to gross primary production, which was not measured. However, we have made attempts to estimate gross production, but this has been done in another paper (Spillling et al 2016) also submitted to the same special issue. We have deleted the text that the reviewer is referring to, and have added a reference to this other paper.

**Editor: Reviewer 1, comment 3**
Not clear which one good point you refer to in first sentence of your response. Reviewer made more than one good point.

**Author response:** The main point is that the UV light is different inside the mesocosm bags compared to the primary production incubations that were carried out in glass vials moored outside the bags. The difference in transmission of UV light may produce a bias, which does not only relate to the primary production measurements but also have other effects such as the release of dissolved organic carbon (DOC) by phytoplankton. These indirect effects were also pointed out by the reviewer and we have taken up these additional points. The reviewer also pointed out that our production and respiration measurements were within values measured before. This is, however, a relatively wide range and we cannot draw any conclusions about any underestimation of the primary production based on that.

We have argued that we think it is a good point, and we will discuss it, but due to the special features of the Baltic Sea with relatively high natural DOC concentrations, which effectively absorb in the UV region, we do not think that UV would have substantially biased the primary production measurements.

We have added a new paragraph about the potential UV effect in the discussion:

Another factor that could have influenced our primary production incubations is UV light, which is a known inhibitor of primary production (Vincent and Roy, 1993), and elevated $CO_2$ concentration may increase the sensitivity to UV light (Sobrino et al., 2009). Additionally, UV light reduces the release of DOC by phytoplankton, in particular at high $CO_2$ concentration (Sobrino et al., 2014), but also cause photochemical mineralization of dissolved organic matter (DOM) (Vahatalo and Jarvinen, 2007). Both DOC release and DOM break down may have implications for bacterial production and nutrient cycling. The mesocosm bags were made in a material absorbing UV light (thermoplastic polyurethane) whereas our primary production incubations were done in glass vials (transmitting some UV light) moored outside the mesocosm bags. The difference in UV transmittance could have produced a bias in the primary production measurements. However, the DOC concentration in the Baltic Sea is very high compared with most other oceans and coastal seas (Hoikkala et al., 2015). Most of this is terrestrial derived, refractory DOC, which effectively absorbs in the UV region, and typically the depth at which 1% of UVB remains is <50 cm (Piazena and Häder, 1994). UVA penetrates a little deeper and may have affected slightly the incubation platform moored at 2 m depth, but we do not believe that UV light caused major inhibition of our primary production measurements or affected phytoplankton DOC production.

**Editor: Reviewer 1, specific comments**
More detail is needed as to what the appropriate changes that you refer to in the last sentence of your response

Author response: There were two additional points made:

Firstly, to add the difference in UV light inside and outside the mesocosm bags to page 20 line 5. We made the changes accordingly (new text in red):

"This suggests a different availability of inorganic nutrients and different plankton community as other environmental variables such as light and temperature were similar both inside and outside the mesocosm bags, except that UV light was absent inside the mesocosm bags."

Secondly, to add a concluding paragraph. We have now added:

"In conclusion, this study suggests that elevated $CO_2$ reduced respiration which in turn increased net carbon fixation. However, the increased primary production did not translate into increased carbon export, and did consequently not work as a negative feedback mechanism for increasing atmospheric $CO_2$ concentration."

**Editor: Reviewer 2, general response**
  provide more detailed response to the comments… I am particularly referring to Reviewer 1, general response last sentence…

Author response: We think that the reviewer has misunderstood some aspects of the uncertainty that we tried to explain. There will always be some measurement uncertainty, which will vary according to e.g. method or instrumentation used. This is not something specific to this study. In the last sentence of the general comments he/she focuses on the uncertainty of TPC measurements, but there is nothing in particular that would indicate that these numbers have an uncertainty that need special consideration, and it does not affect the main conclusions of the paper. The data not normalized to TPC (which is one of the concerns) was presented in the supplementary material originally submitted. From these it is evident that the normalization to TPC does not change the main conclusion that respiration decreased at the high $CO_2$ treatment.

**Editor: Reviewer 2, comment 4**
address also the comment on presentation of the data

Author response. The reviewer suggested to change the stacked bar plots into stacked area plots and show the full time series of data. Although the community composition data is important, we do not think it is within the scope of the paper to present the whole time series data of community development, in particular since it is the main focus of two other papers in the special issue: Lischka et al. for the zooplankton and Bermudez et al. for the phytoplankton community. To make this clear we have included references to these papers in the figure legend.

**Editor: Reviewer 2, comment 5**
response is not complete; miss response to comment to include virus induced C losses

This is a bit outside the scope of the paper because there are other papers in the special issue that deals with this topic (Crawfurd et al, and the other Spilling et al), but we have added some more information about this in the discussion:

Temporal changes in bacterial abundances followed largely that of phytoplankton biomass, and there were significant increases in viral lysis rates in the high $CO_2$ treatment (Crawfurd et al., 2016). This was most likely a consequence of higher abundances of pico-eukaryotes and pointing towards a more productive but regenerative system (Crawfurd et al., 2016).

**Our original response with additional information of the concrete changes made in the manuscript**

**Dr. Neale**

**Reviewer #1, comment #1.** While the results do demonstrate decreased respiration for samples from the higher CO2 enrichments, I do have some concern about how representative these rates are of processes in the mesocosms. A depth integrated water sample was taken and incubated at "ambient" temperature. But it can be seen from Paul et al (2015) that there was a strong temperature gradient over the mesocosm's depth range, at times as much as 10_C, so it is not clear what was "ambient" temperature. Moreover, mixing waters of differing temperatures may bias the respiration measurement at a fixed temperature vs. the "real" average, i.e. combining warm, lower particle concentration surface water with cooler, high particle (or nutrient) concentration bottom water could stimulate respiration versus the average of the two.

Author response:

It is true that temperature stratification varied. We kept the incubation temperature at the surface temperature, and we will add this information to the Materials and Methods chapter. Dr. Neale makes a valid point that there might be a bias due to mixing water of different temperature rather than averaging measurements taken at different temperature. Logistical constraints prevented us from making respiration incubations at several temperatures. We will take up this potential bias in the Discussion chapter.

Changes made:

In the materials and methods chapter we added the information that we were using the surface temperature (changes made in red):

"After the initial $O_2$ determination, the bottles were put in a dark, temperature controlled room, set to the ambient water temperature at the surface."

In the discussion we added a new paragraph (under section 4.2) about the potential bias:

"Having the respiration incubation at a fixed temperature might have caused a slight bias as there was varying thermal stratification throughout the experiment and the temperature was not even throughout the mesocosm bags. A better approach would have been to have respiration incubations in temperatures above and below the thermocline, but logistical constrains prevented us from doing this."

**Reviewer #1, comment #2.** The authors also indicate that respired carbon was about 10x greater than net production (pg. 17 line 7). Some more explanation is needed for why such comparison is made since a determination of whether the system is net heterotrophic or autotrophic would require comparison of gross primary production with total community respiration, as stated on page 21 line 9. The statement on page 21 line 26 implies that the authors have some idea of gross primary production, could this be compared to respiration rate?

Author response:

We did try to estimate the gross primary production and after the submission of this paper we made a carbon budget for the whole experiment, which is presented in a synthesis paper (Spilling et al 2016). We will remove these statements from the present paper and place a reference to the budget paper in order to have a more clear focus.

Changes made:
This section was removed from the first paragraph in section 4.2.

**Reviewer #1, comment #3.** The authors also speculate that the net primary productivity method may not have been sensitive enough to detect difference between treatments, so that enhanced production at increased CO2 was not detected. Small incubation volumes are suggested to contribute to uncertainty but the authors give no indication of what was that measurement uncertainty. Nevertheless, they state that the measurements were comparable to previous ones in the same regions using similar methods (Kivi et al. 1993) which would argue against any substantial bias. One other factor to consider as to whether the NPP assay would detect an enhancement effect was that the incubations were conducted outside the bags. According to Riebesell et al. (2013), the mesocosm material (thermoplastic polyurethane) removes all UV whereas glass scintillation vials used for the NPP incubation transmit UV-A and most UV-B so rates in the vials could have been substantially more inhibited in the near surface samples than phytoplankton in the mesocosms that were protected from UV. Moreover, some studies have shown that phytoplankton grown under CO2 enhanced conditions are more sensitive to UV. It is possible that NPP was higher in the mesocosms with CO2 enrichment but the effect was dampened in incubations outside the bag due to a counterbalancing increase in sensitivity to UV (see, e.g., Sobrino et al. 2008, 2009). Also, as the lead author knows (since he was co-author on the paper), Sobrino et al. (2014) observed lower rates of DOC release during short term PPR incubations by phytoplankton acclimated to CO2 enhanced conditions but this effect was much less when incubations included UV. This DOC would be quite labile and rapidly respired so might not affect the bulk DOC pool but a reduction in DOC release could decrease bacterial respiration.

Author response:
A very good point that we will take up in the Discussion chapter. The DOC concentration in the Baltic Sea is very high compared with most other oceans and coastal seas (like the Mediterranean that is referred to). Most of this is refractory DOC, which effectively absorbs in the UV region, and typically the depth at which 1% of UVB remains is <50 cm (e.g. Piazena and Häder 1994). UVA penetrates a little deeper and may have affected slightly the incubation platform moored at 2 m depth. We do not believe, however, that UV light have caused major inhibition of our primary production measurements (or affected labile DOC production), but we will point this out with the reasoning described above.

Changes made:
We have added a new paragraph about the potential UV effect in the discussion:

"Another factor that could have influenced our primary production incubations is UV light, which is a known inhibitor of primary production (Vincent and Roy, 1993), and elevated $CO_2$ concentration may increase the sensitivity to UV light (Sobrino et al., 2009). Additionally, UV light reduces the release of DOC by phytoplankton, in particular at high $CO_2$ concentration (Sobrino et al., 2014), but also cause photochemical mineralization of dissolved organic matter (DOM) (Vahatalo and Jarvinen, 2007). Both DOC release and DOM break down may have implications for bacterial production and nutrient cycling. The mesocosm bags were made in a material absorbing UV light (thermoplastic polyurethane) whereas our primary production incubations were done in glass vials (transmitting some UV light) moored outside the mesocosm bags. The difference in UV transmittance could have produced a bias in the primary production measurements. However, the DOC concentration in the Baltic Sea is very high compared with most oceans and coastal seas (Hoikkala et al., 2015). Most of this is terrestrial derived, refractory DOC, which effectively absorbs in the UV region, and typically the depth at which 1% of UVB remains is <50 cm (Piazena and Häder, 1994). UVA penetrates a little deeper and may have affected slightly the incubation platform moored at 2 m depth, but we do not believe that UV light caused major inhibition of our primary production measurements or affected phytoplankton DOC production. "

.

**Reviewer #2**

**Reviewer #2, comment #1.** I have several concerns that (in my opinion) warrant further attention from the authors. I found surprising the lack of real independent mesocosms replicates. Only the controls do replicate (M1 and M5). Under these circumstances an appropriate statistical analysis cannot be performed, compromising the significance of the results. In its place, regression coefficient significance tests have been done to analyse the significance of the results. Although valid, these tests compare the mesocosms between them, but the behaviour, obviously implying variability within each specific treatment cannot then be ruled out, because without replicates is not possible to discern if the response is due to the controlled factor (CO2) or to any other uncontrolled factor, and or their interaction. At least, significance differences obtained from the R comparisons tests should be mentioned in the text adding the p values (in results section) and marked in the Figures as an asterisk or letter to indeed demonstrate that there are some differences. A table including the results of all linear regression analyses indicating the significant effects of the different CO2 concentrations on the variables would needed (see Tables and Figures in Paul et al. 2015, Crawfurd et al., 2015, Bermudez et al., 2015-this special issue-as examples of what I am referring to). In my opinion, in this manner the Ms would benefit of a better understanding of the results.

Author response:
The mesocosm bags are relatively large scale operations, 55 $m^3$ in each enclosure, and this puts some constrains on how many units can be used. Lack of replication does not, however, prevent proper statistical analysis of results: for example gradient experiments of a single variable or factorial design experiments with multiple variables, both provide data that can be statistically tested for treatment effects (see e.g. the discussion by Oksanen 2001 and Hurlbert 2004). In our case, a gradient of different $CO_2$ additions was used. The statistical test was in the figure legend, and we will include it also in the results section as suggested by the reviewer.

Changes made: We added the details to the text in the results section slope -0.0002; p = 0.02; $R^2$ = 0.77;

**Reviewer #2, comment #2.** The other important issue is that you mention measurement uncertainties at some points. I do not understand how or why can be a measurement uncertainty working with small volumes, can you specify? How this affect reliability? Regarding the incubation time with 14C, I think it is widely demonstrated that this method is quite sensitive. I agree that it may be more estimative of NPP, but, in incubations long as 24h, the same 14C molecule can be fixed and respired several times (the eternal discussion). Do you think you could be getting an underestimate of your measurement? Said this, I think the point raised by reviewer 1 regarding the effect of UV on C fixation during incubations would be much more relevant in terms of affecting PP (not commenting on UVR as I totally agree and support reviewer 1 comments). Also said by reviewer 1, if you think there are uncertainties, how your data compare to former published studies?

Author response:
There will always be measurements uncertainties (depending on the methodology, instrument etc) and this would be independent of the volume, and we are not quite sure that we got your point. Perhaps you refer to the primary production measurements. In that case the incubation volume was relatively small, and we did not remove the grazers, which could have introduced a bias with respect to grazing pressure, i.e. the number of grazers could have been quite variable depending on how many by chance got into the relatively small incubation volume.

With respect to the UV point, please see our response above to reviewer #1.

**Reviewer #2, comment #3.** It is not clear to me whether you also mention measurement uncertainties on the TPC data, it seems so. In this regards, if there exist such an uncertainty in TPC, how this translates into figures 4 and 5 that are normalised by TPC? The -under or –sub estimations would then be included in your calculations on the cumulative PP and TR
and vertical C flux?

Author response:
There are of course also measurements uncertainties in the TPC, and yes these would be included in the data presented in Figs 4 and 5. However, it does not affect the main conclusion of the paper.

Additional author comments
Please see also our comments made to the editor regarding this point

**Reviewer #2, comment #4.** Phytoplankton community composition. As data are presented it is not clearly seen that there is dominance of some groups over others. Only Euglenophytes seem to be absent in t0. Dinoflagellates, cyanobacteria, diatoms and chlorophytes look like having similar proportions in t0 and t17 (p values needed), while "other" increase at 1333 ppm. What organisms does "other" comprise? Stacked area plots would give a much better idea of the temporal evolution and trend followed by the community and so significances could be better appreciated. Thus I suggest to re-plot figures 1 and 2 including all days and treatments in stacked area charts. How your data compare to Bermudez et al. this issue-seems that taxonomy differs a little in between the two studies (for instance Euglenophyta).

Author response:
We wanted to present a general overview of the plankton community composition, and a more in-depth analysis and presentation of all the dates are provided in Bermudez et al 2016 and Lischka et al 2015. The presented phytoplankton data is the same as Bermudez et al 2016, but here we have additionally included counts of phytoplankton >20μm, affecting the biomass of e.g. Euglenophyta.

Additional author comments
Please see also our comments made to the editor regarding this point

**Reviewer #2, comment #5.** Considering that your study deals with the plankton food web, bacterial production, or at least abundances have not been analysed. Although probably low in volume and biomass contribution as compared to phyto and zooplankton groups, they are important too since they have been reported to react positively to increased $CO_2$ (a number of papers published on this topic by Grossart, Schulz and Riebesell). I see bacterial contribution is further discussed in pg. 20 based on former reports. How about bacterial production/abundances in this very mesocosms experiment? Neither you say anything about viruses affecting C losses, which is important for C cycling and definitively affect C export. These two (bacteria and viruses) in my opinion shall be at least being discussed (succinctly if you wish) within the framework of the whole mesocosm experiment.

Author response:

A very good point and we will incorporate this into the discussion. Bacterial production was measured (Hornick et al 2016), but this was not out yet at the time of review. Heterotrophic prokaryotes were enumerated and this data is presented in Crawfurd et al (2016).

Additional author comments and changes made

This is a bit outside the scope of the paper because there are other papers in the special issue that deals with this topic (Crawfurd et al, and the other Spilling et al), but we have added some more information about this in the discussion:

Temporal changes in bacterial abundances followed largely that of phytoplankton biomass, and there were significant increases in viral lysis rates in the high $CO_2$ treatment (Crawfurd et al., 2016). This was most likely a consequence of higher abundances of pico-eukaryotes and pointing towards a more productive but regenerative system (Crawfurd et al., 2016).

**Reviewer #2, comment #6.** Pg. 18. Ln 20. "The larger-scale mesocosms … interacting effects between different components of the food web are included". Pg. 19 Ln 21. Subheading "Interacting effects and community composition". Also in pg. 20 Ln 10 interactive effects are mentioned. I find this an overstatement since you have not analysed interactive effects

Author response:

This section is under the discussion of advantages of mesocosm experiments on a general level. We will change this to "possible interacting effects…" to make it more clear.

Changes made:

Added 'possible' to this sentence.

**Reviewer #2, comment #7.** Pg. 18 Ln 22-pg. 19 Ln 6. Instead of discussing higher plants which do not deal with carbonic /carbonate equilibrium and the systems are different, I think it would make much more sense to focus on explaining the mechanisms why respiration might be reduced in aquatic organisms such as phytoplankton at high CO2. Can the decreased TR be related to CCMs? Both photosynthesis and respiration generate energy that can be used for CCMs since they are mechanisms highly-energy - demanding. Under increased CO2 it is well known that CCMs are downregulated. If there are no active CCMs, then respiration and photosynthesis might also be downregulated, and the energy consumed by them is "available" for other purposes. On the other hand, such energy could also be directed to growth (i.e. PP) that is what you are describing. This would mean that respiration could be downregulated but not PP. Such uncoupling is what is important to discuss in depth. Also, how is this related to pigment concentration? Since under high CO2 there is less electronic demand, pigments should decrease. Indeed Chla sharply decreased from 2 ugL-1 on P1 to 0.8 in PII and III (Paul et al., 2015). However you estate in pg. 22 Ln 6 that CO2 had a positive effect on Chla. Some clarification is needed.

Author response:

A very good point and a more thorough discussion around CCMs will be incorporated into the Discussion chapter. As for the Chl a, the major change was driven by change in total phytoplankton biomass, e.g. the overall decrease from PI to PII and PIII, and the higher Chl a in the high $CO_2$.

Additional comments and changes made

Here we would like to add that higher plants are more studied with respect to changes in $CO_2$ and this literature is relevant in terms of e.g. enzymatic activity. That said, the underlying processes are not well understood and we are not able to pin point the exact mechanisms that could cause reduced respiration. We have incorporated the references that the reviewer suggested and also added a paragraph on possible effects of CCM:

Changes in carbonate chemistry speciation might also affect the availability of the sole substrate, i.e. $CO_2$, at the site of photosynthetic carbon fixation. At present, marine waters typically have a pH of 8 or above, and most of the carbon is in the form of bicarbonate ($HCO_3^-$). Many phytoplankton groups have developed carbon concentrating mechanisms (CCMs), such as the active uptake of bicarbonate, as a way to increase substrate availability at the site of carbon fixation (Singh et al., 2014). Increased CO2 availability may reduce metabolic activity related to CCMs, which would affect the respiration rate of primary producers.

**Reviwer #2, Specific comments and technical suggestions**

Author repose
We will make appropriate changes to all the specific comments and technical suggestions raised by the reviewer.

Changes made:
We have added a concluding paragraph as mentioned under reviewer #1, and changed the term 'parameter' to 'variable' throughout the text.

Additional comment and changes

Thanks to a two helpful references provided by reviewer #2 and his/her comment about expanding on the discussion around the effect of pH on respiration we expanded a bit on the discussion paragraph about this topic:

"However, this is not straight forward and more studies of the effect of changed external pH on membrane transport are needed (Taylor et al., 2012). There might additionally be considerable difference between marine organisms depending on e.g. size, metabolic activity and growth rates, which directly affect pH in the diffusive boundary layer surrounding the organism (Flynn et al., 2012). "

**Other changes made to the manuscript**

We have also made slight modifications to Table 1, after seeing the review of the synthesis paper submitted to the special issue (Spilling et al 2016). We changed the exported carbon parameters as the original data included sampling at T-1 when the CO2 concentration had not yet been altered. We deleted the ΔTPC parameter as this is presented in the synthesis paper, and not really relevant for this paper, and we changed the TPC pool to be an average for the two periods. We also added Standard Error for all calculations, with an addition to the legend how this was done, and to the Materials and Methods under section 2.9 (last paragraph):

"From the two different phases of the experiment (Phases I and II; *t0 – t16* and *t17 – t31* respectively) we calculated the average for the different parameters and SE, with 9 and 7 sampling points during Phase I and II respectively."

Modified text highlighted in yellow, please also see the cover letter for a detailed description of the changes made to the manuscript

[revised manuscript text omitted]

---

## Author Response (AR2)

**Response to second review**

Dear Editor

We have gone through the second revision, please find our comments to the points below.

Yours sincerely,

Kristian Spilling

Reviewer comments on MS No.: bg-2015-608

**Reviewer comment #1**

2nd REVISION: The revised version of the manuscript by Spilling et al. addresses both reviewer's comments in a satisfactorily manner, except for one point that I still think disserves more attention. Although I consider it a minor change, it shall be properly addressed.

1. Ln480-491. I do not see the rationale of still including discussion on vascular plants, despite of authors arguments in favour of doing it. For most of the phytoplanktonic species, rubisco is less than half-saturated under present $CO_2$ concentrations in seawater and, that is the point, some species have developed ways to increase intracellular $CO_2$ concentrations by using CCMs, possibly having a direct relationship with respiration. I insist, vascular plants behave in a very different manner than phytoplankton due to the particularity of the aquatic environment.

I do not agree with this sentence: "Phytoplankton lacks any specialized structures like root system and may consequently function more like plant foliage", which seems to be the "excuse" for not having removed the paragraph related to vascular plants. I think that suggesting that "phytoplankton may act as plant foliage" is going a bit too far...

Thus, as I said in my former report, mentioning embryophytes here is out place, when there are very relevant studies done in CCMs in phytoplankton that could have been used to better contextualize this

part of the discussion. Remove the mentioned-paragraph, and discuss your data attending to phytoplankton CCMs.

**Author response:**

We disagree somewhat on this topic, but this section is not critical for the discussion and we removed the paragraph in question.

We moved up, and expanded on the CCMs replacing the paragraph the reviewer pointed out (see also comment #2). The modified paragraph now reads:

"For primary producers in aquatic environment, changes in carbonate chemistry speciation affects the availability of the sole substrate, i.e. $CO_2$, at the site of photosynthetic carbon fixation. At present, marine waters typically have a pH of 8 or above, and most of the carbon is in the form of bicarbonate ($HCO_3^-$). Many phytoplankton groups have developed carbon concentrating mechanisms (CCMs) as a way to increase substrate availability at the site of carbon fixation (Singh et al., 2014), reducing the cost of growth (Raven, 1991). For phytoplankton with CCMs, increased $CO_2$ availability would suppress the CCM, freeing resources for growth, in particular under light limiting conditions (Beardall and Giordano, 2002). There are examples of experiments with ocean acidification that has indicated downregulation of CCM (Hopkinson et al., 2010) and photosynthetic apparatus (Sobrino et al., 2014), which could reduce respiration in phytoplankton."

**Reviewer comment #2**

Check the following refs please:

-Badger, M.R., Andrews, T.J., Whitney, S.M., Ludwig, M., Yellowlees, D.C., Leggat, W., and Price, G.D. (1998) The diversity and coevolution of Rubisco, plastids, pyrenoids, and chloroplast-based CO2 - concentrating mechanisms in algae. Can. J. Bot. 76: 1052–1071.

-Beardall, J. and Giordano, M. (2002) Ecological implications of microalgal and cyanobacterial CO2 concentrating mechanisms, and their regulation. Funct. Plant Biol. 29: 335–347.

-Emma Huertas, Brian Colman, and George S. Espie (2002). Mitochondrial-Driven Bicarbonate Transport Supports Photosynthesis in a Marine Microalga . Plant Physiol.130:284-291

-Nimer, N.A., Merrett, M.J., and Brownlee, C. (1996) Inorganic carbon transport in relation to culture age and inorganic carbon concentration in a high-calcifying strain of Emiliania huxleyi (Prymnesiophyceae). J. Phycol. 32: 813.818

-Raven, J.A. (1991) Physiology of inorganic C acquisition and implications for resource use efficiency by marine phytoplankton: relation to increased CO2 and temperature. Plant, Cell Environ. 14: 779–794

-Sültemeyer, D. (1998) Carbonic anhydrase in eukaryotic algae: characterization, regulation, and possible function during photosynthesis. Can. J. Bot. 76: 962–972

**Author response**: we have gone through the references and has expanded on the discussion on CCMs. We added references to Beardall and Giordano (2002) and Raven (1991)

**Reviewer comment #3**

2. Supplemental Figure 1S (very minor comment) showing the platforms used for primary production incubations does not read well. The font used to depict the different depths, buoys anchors, sediments traps etc. would do better in bold bigger size font on top of the green colour representing the water column.

**Author response**: We made a new version of the figure with larger text.